# Representation of internal speech by single neurons in human supramarginal gyrus

Sarah K. Wandelt ●[1,2] ✉, David A. Bjånes[1,2,3], Kelsie Pejsa[1,2], Brian Lee[1,4,5], Charles Liu ●[1,3,4,5] & Richard A. Andersen[1,2]

Speech brain–machine interfaces (BMIs) translate brain signals into words or audio outputs, enabling communication for people having lost their speech abilities due to diseases or injury. While important advances in vocalized, attempted and mimed speech decoding have been achieved, results for internal speech decoding are sparse and have yet to achieve high functionality. Notably, it is still unclear from which brain areas internal speech can be decoded. Here two participants with tetraplegia with implanted microelectrode arrays located in the supramarginal gyrus (SMG) and primary somatosensory cortex (S1) performed internal and vocalized speech of six words and two pseudowords. In both participants, we found significant neural representation of internal and vocalized speech, at the single neuron and population level in the SMG. From recorded population activity in the SMG, the internally spoken and vocalized words were significantly decodable. In an offline analysis, we achieved average decoding accuracies of 55% and 24% for each participant, respectively (chance level 12.5%), and during an online internal speech BMI task, we averaged 79% and 23% accuracy, respectively. Evidence of shared neural representations between internal speech, word reading and vocalized speech processes was found in participant 1. SMG represented words as well as pseudowords, providing evidence for phonetic encoding. Furthermore, our decoder achieved high classification with multiple internal speech strategies (auditory imagination/visual imagination). Activity in S1 was modulated by vocalized but not internal speech in both participants, suggesting no articulator movements of the vocal tract occurred during internal speech production. This work represents a proof-of-concept for a high-performance internal speech BMI.

Speech is one of the most basic forms of human communication, a natural and intuitive way for humans to express their thoughts and desires. Neurological diseases like amyotrophic lateral sclerosis (ALS) and brain lesions can lead to the loss of this ability. In the most severe cases, patients who experience full-body paralysis might be left without any means of communication. Patients with ALS self-report loss of speech as their most serious concern[1]. Brain–machine interfaces (BMIs) are devices offering a promising technological path to bypass

[1]Division of Biology and Biological Engineering, California Institute of Technology, Pasadena, CA, USA. [2]T&C Chen Brain-Machine Interface Center, California Institute of Technology, Pasadena, CA, USA. [3]Rancho Los Amigos National Rehabilitation Center, Downey, CA, USA. [4]Department of Neurological Surgery, Keck School of Medicine of USC, Los Angeles, CA, USA. [5]USC Neurorestoration Center, Keck School of Medicine of USC, Los Angeles, CA, USA. ✉e-mail: skwandelt@gmail.com

neurological impairment by recording neural activity directly from the cortex. Cognitive BMIs have demonstrated potential to restore independence to participants with tetraplegia by reading out movement intent directly from the brain[2–5]. Similarly, reading out internal (also reported as inner, imagined or covert) speech signals could allow the restoration of communication to people who have lost it.

Decoding speech signals directly from the brain presents its own unique challenges. While non-invasive recording methods such as functional magnetic resonance imaging (fMRI), electroencephalography (EEG) or magnetoencephalography[6] are important tools to locate speech and internal speech production, they lack the necessary temporal and spatial resolution, adequate signal-to-noise ratio or portability for building an online speech BMI[7–9]. For example, state-of-the-art EEG-based imagined speech decoding performances in 2022 ranged from approximately 60% to 80% binary classification[10]. Intracortical electrophysiological recordings have higher signal-to-noise ratios and excellent temporal resolution[11] and are a more suitable choice for an internal speech decoding device.

Invasive speech decoding has predominantly been attempted with electrocorticography (ECoG)[9] or stereo-electroencephalographic depth arrays[12], as they allow sampling neural activity from different parts of the brain simultaneously. Impressive results in vocalized and attempted speech decoding and reconstruction have been achieved using these techniques[13–18]. However, vocalized speech has also been decoded from localized regions of the cortex. In 2009, the use of a neurotrophic electrode[19] demonstrated real-time speech synthesis from the motor cortex. More recently, speech neuroprosthetics were built from small-scale microelectrode arrays located in the motor cortex[20,21], premotor cortex[22] and supramarginal gyrus (SMG)[23], demonstrating that vocalized speech BMIs can be built using neural signals from localized regions of cortex.

While important advances in vocalized speech[16], attempted speech[18] and mimed speech[17,22,24–26] decoding have been made, highly accurate internal speech decoding has not been achieved. Lack of behavioural output, lower signal-to-noise ratio and differences in cortical activations compared with vocalized speech are speculated to contribute to lower classification accuracies of internal speech[7,8,13,27,28]. In ref. 29, patients implanted with ECoG grids over frontal, parietal and temporal regions silently read or vocalized written words from a screen. They significantly decoded vowels (37.5%) and consonants (36.3%) from internal speech (chance level 25%). Ikeda et al.[30] decoded three internally spoken vowels using ECoG arrays using frequencies in the beta band, with up to 55.6% accuracy from the Broca area (chance level 33%). Using the same recording technology, ref. 31 investigated the decoding of six words during internal speech. The authors demonstrated an average pair-wise classification accuracy of 58%, reaching 88% for the highest pair (chance level 50%). These studies were so-called open-loop experiments, in which the data were analysed offline after acquisition. A recent paper demonstrated real-time (closed-loop) speech decoding using stereotactic depth electrodes[32]. The results were encouraging as internal speech could be detected; however, the reconstructed audio was not discernable and required audible speech to train the decoding model.

While, to our knowledge, internal speech has not previously been decoded from SMG, evidence for internal speech representation in the SMG exists. A review of 100 fMRI studies[33] not only described SMG activity during speech production but also suggested its involvement in subvocal speech[34,35]. Similarly, an ECoG study identified high-frequency SMG modulation during vocalized and internal speech[36]. Additionally, fMRI studies have demonstrated SMG involvement in phonologic processing, for instance, during tasks while participants reported whether two words rhyme[37]. Performing such tasks requires the participant to internally 'hear' the word, indicating potential internal speech representation[38]. Furthermore, a study performed in people suffering from aphasia found that lesions in the SMG and its adjacent white matter

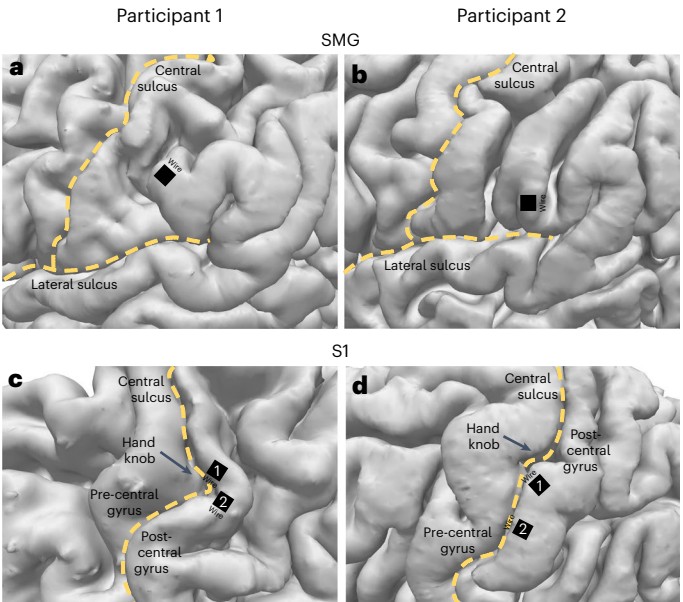

**Fig. 1 | Multielectrode implant locations. a,b**, SMG implant locations in participant 1 (1 × 96 multielectrode array) (**a**) and participant 2 (1 × 64 multielectrode array) (**b**). **c,d**, S1 implant locations in participant 1 (2 × 96 multielectrode arrays) (**c**) and participant 2 (2 × 64 multielectrode arrays) (**d**).

affected inner speech rhyming tasks[39]. Recently, ref. 16 showed that electrode grids over SMG contributed to vocalized speech decoding. Finally, vocalized grasps and colour words were decodable from SMG from one of the same participants involved in this work[23]. These studies provide evidence for the possibility of an internal speech decoder from neural activity in the SMG.

The relationship between inner speech and vocalized speech is still debated. The general consensus posits similarities between internal and vocalized speech processes[36], but the degree of overlap is not well understood[8,35,40–42]. Characterizing similarities between vocalized and internal speech could provide evidence that results found with vocalized speech could translate to internal speech. However, such a relationship may not be guaranteed. For instance, some brain areas involved in vocalized speech might be poor candidates for internal speech decoding.

In this Article, two participants with tetraplegia performed internal and vocalized speech of eight words while neurophysiological responses were captured from two implant sites. To investigate neural semantic and phonetic representation, the words were composed of six lexical words and two pseudowords (words that mimic real words without semantic meaning). We examined representations of various language processes at the single-neuron level using recording microelectrode arrays from the SMG located in the posterior parietal cortex (PPC) and the arm and/or hand regions of the primary somatosensory cortex (S1). S1 served as a control for movement, due to emerging evidence of its activation beyond defined regions of interest[43,44]. Words were presented with an auditory or a written cue and were produced internally as well as orally. We hypothesized that SMG and S1 activity would modulate during vocalized speech and that SMG activity would modulate during internal speech. Shared representation between internal speech, vocalized speech, auditory comprehension and word reading processes was investigated.

## Results
### Task design
We characterized neural representations of four different language processes within a population of SMG and S1 neurons: auditory

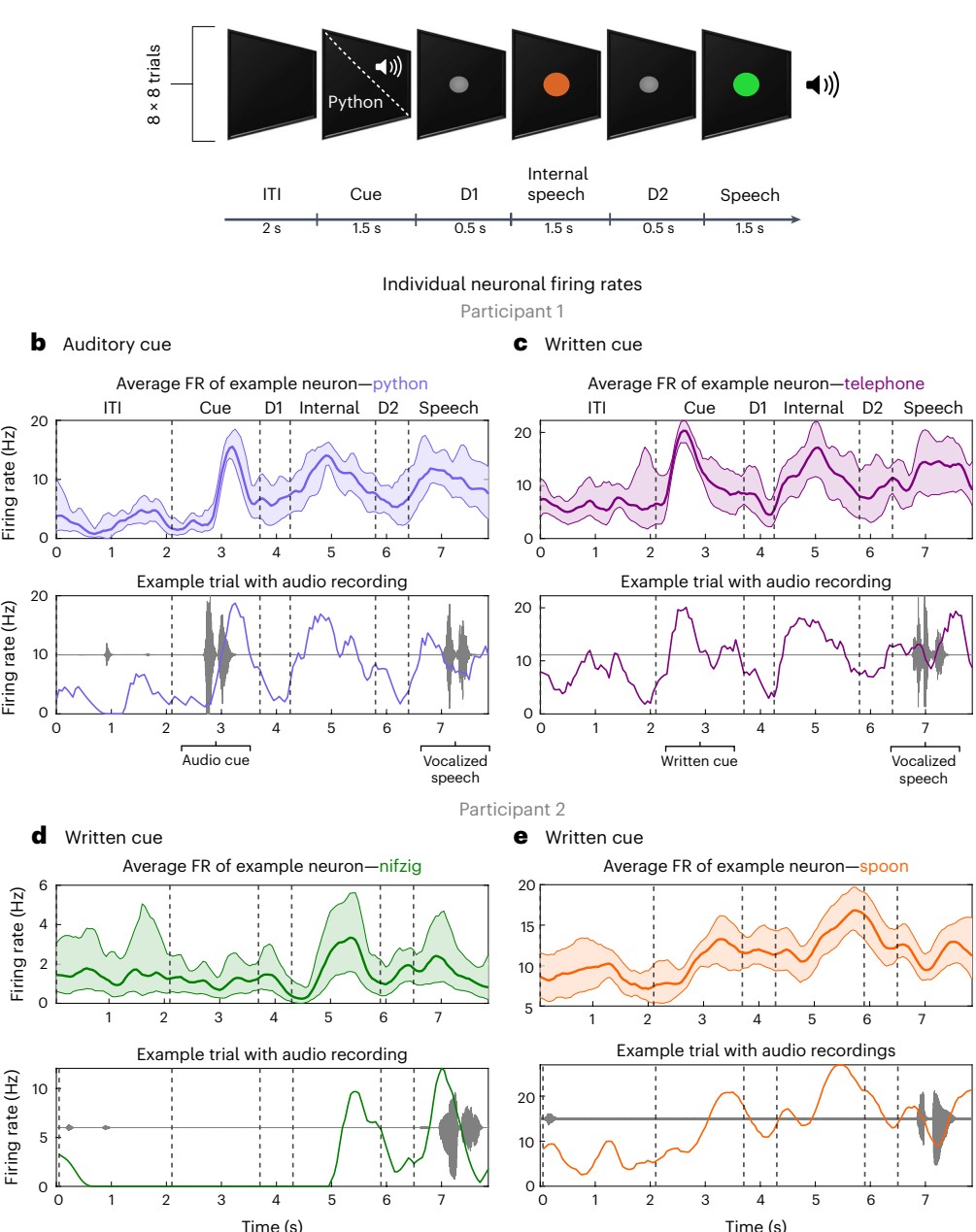

**a** Internal speech task design

Individual neuronal firing rates
Participant 1

**b** Auditory cue

Average FR of example neuron—python

**c** Written cue

Average FR of example neuron—telephone

Participant 2

**d** Written cue

Average FR of example neuron—nifzig

**e** Written cue

Average FR of example neuron—spoon

**Fig. 2 | Neurons in the SMG represent language processes. a**, Written words and sounds were used to cue six words and two pseudowords in a participant with tetraplegia. The 'audio cue' task was composed of an ITI, a cue phase during which the sound of one of the words was emitted from a speaker (between 842 and 1,130 ms), a first delay (D1), an internal speech phase, a second delay (D2) and a vocalized speech phase. The 'written cue' task was identical to the 'audio cue' task, except that written words appeared on the screen for 1.5 s. Eight repetitions of eight words were performed per session day and per task for the first participant. For the second participant, 16 repetitions of eight words were performed for the written cue task. **b–e**, Example smoothed firing rates of neurons tuned to four words in the SMG for participant 1 (auditory cue, python (**b**), and written cue, telephone (**c**)) and participant 2 (written cue, nifzig (**d**), and written cue, spoon (**e**)). Top: the average firing rate over 8 or 16 trials (solid line, mean; shaded area, 95% bootstrapped confidence interval). Bottom: one example trial with associated audio amplitude (grey). Vertically dashed lines indicate the beginning of each phase. Single neurons modulate firing rate during internal speech in the SMG.

comprehension, word reading, internal speech and vocalized speech production. In this manuscript, internal speech refers to engaging a prompted word internally ('inner monologue'), without correlated motor output, while vocalized speech refers to audibly vocalizing a prompted word. Participants were implanted in the SMG and S1 on the basis of grasp localization fMRI tasks (Fig. 1).

The task contained six phases: an inter-trial interval (ITI), a cue phase (cue), a first delay (D1), an internal speech phase (internal), a second delay (D2) and a vocalized speech phase (speech). Words were cued with either an auditory or a written version of the word (Fig. 2a). Six of the words were informed by ref. 31 (battlefield, cowboy, python, spoon, swimming and telephone). Two pseudowords (nifzig and bindip) were added to explore phonetic representation in the SMG. The first participant completed ten session days, composed of both the auditory and the written cue tasks. The second participant completed nine sessions, focusing only on the written cue task. The participants

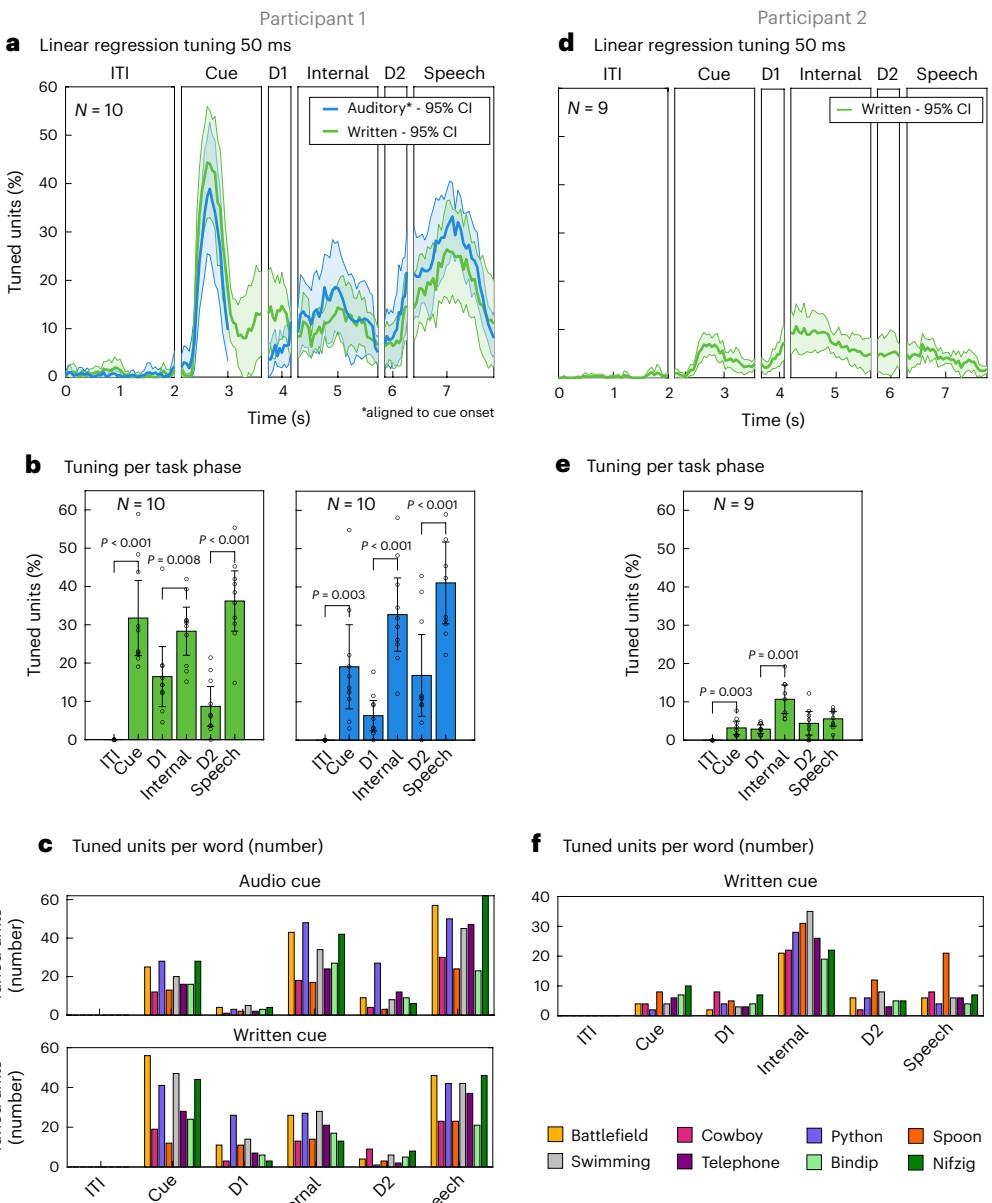

**Fig. 3 | Neuronal population activity modulates for individual words. a**, The average percentage of tuned neurons to words in 50-ms time bins in the SMG over the trial duration for 'auditory cue' (blue) and 'written cue' (green) tasks for participant 1 (solid line, mean over ten sessions; shaded area, 95% confidence interval of the mean). During the cue phase of auditory trials, neural data were aligned to audio onset, which occurred within 200–650 ms following initiation of the cue phase. **b**, The average percentage of tuned neurons computed on firing rates per task phase, with 95% confidence interval over ten sessions. Tuning during action phases (cue, internal and speech) following rest phases (ITI, D1 and D2) was significantly higher (paired two-tailed *t*-test, d.f. 9, $P_{ITI\_CueWritten} < 0.001$, Cohen's $d = 2.31$; $P_{ITI\_CueAuditory} = 0.003$, Cohen's $d = 1.25$; $P_{D1\_InternalWritten} = 0.008$, Cohen's $d = 1.08$; $P_{D1\_InternalAuditory} < 0.001$, Cohen's $d = 1.71$; $P_{D2\_SpeechWritten} < 0.001$,

Cohen's $d = 2.34$; $P_{D2\_SpeechAuditory} < 0.001$, Cohen's $d = 3.23$). **c**, The number of neurons tuned to each individual word in each phase for the 'auditory cue' and 'written cue' tasks. **d**, The average percentage of tuned neurons to words in 50-ms time bins in the SMG over the trial duration for 'written cue' (green) tasks for participant 2 (solid line, mean over nine sessions; shaded area, 95% confidence interval of the mean). Due to a reduced number of tuned units, only the 'written cue' task variation was performed. **e**, The average percentage of tuned neurons computed on firing rates per task phase, with 95% confidence interval over nine sessions. Tuning during cue and internal phases following rest phases ITI and D1 was significantly higher (paired two-tailed *t*-test, d.f. 8, $P_{ITI\_CueWritten} = 0.003$, Cohen's $d = 1.38$; $P_{D1\_Internal} = 0.001$, Cohen's $d = 1.67$). **f**, The number of neurons tuned to each individual word in each phase for the 'written cue' task.

were instructed to internally say the cued word during the internal speech phase and to vocalize the same word during the speech phase.

For each of the four language processes, we observed selective modulation of individual neurons' firing rates (Fig. 2b–e). In general, the firing rates of neurons increased during the active phases (cue, internal and speech) and decreased during the rest phases (ITI, D1 and D2). A variety of activation patterns were present in the neural population. Example neurons were selected to demonstrate increases

in firing rates during internal speech, cue and vocalized speech. Both the auditory (Fig. 2b) and the written cue (Fig. 2c–e) evoked highly modulated firing rates of individual neurons during internal speech.

These stereotypical activation patterns were evident at the single-trial level (Fig. 2b–e, bottom). When the auditory recording was overlaid with firing rates from a single trial, a heterogeneous neural response was observed (Supplementary Fig. 1a), with some SMG neurons preceding or lagging peak auditory levels during vocalized

speech. In contrast, neural activity from primary sensory cortex (S1) only modulated during vocalized speech and produced similar firing patterns regardless of the vocalized word (Supplementary Fig. 1b).

### Population activity represented selective tuning for individual words

Population analysis in the SMG mirrored single-neuron patterns of activation, showing increases in tuning during the active task phases (Fig. 3a,d). Tuning of a neuron to a word was determined by fitting a linear regression model to the firing rate in 50-ms time bins (Methods). Distinctions between participant 1 and participant 2 were observed. Specifically, participant 1 exhibited strong tuning, whereas the number of tuned units was notably lower in participant 2. Based on these findings, we exclusively ran the written cue task with participant number 2. In participant 1, representation of the auditory cue was lower compared with the written cue (Fig. 3b, cue). However, this difference was not observed for other task phases. In both participants, the tuned population activity in S1 increased during vocalized speech but not during the cue and internal speech phases (Supplementary Fig. 3a,b).

To quantitatively compare activity between phases, we assessed the differential response patterns for individual words by examining the variations in average firing rate across different task phases (Fig. 3b,e). In both participants, tuning during the cue and internal speech phases was significantly higher compared with their preceding rest phases ITI and D1 (paired $t$-test between phases. Participant 1: d.f. 9, $P_{ITI\_CueWritten} < 0.001$, Cohen's $d = 2.31$; $P_{ITI\_CueAuditory} = 0.003$, Cohen's $d = 1.25$; $P_{D1\_InternalWritten} = 0.008$, Cohen's $d = 1.08$; $P_{D1\_InternalAuditory} < 0.001$, Cohen's $d = 1.71$. Participant 2: d.f. 8, $P_{ITI\_CueWritten} = 0.003$, Cohen's $d = 1.38$; $P_{D1\_Internal} = 0.001$, Cohen's $d = 1.67$). For participant 1, we also observed significantly higher tuning to vocalized speech than to tuning in D2 (d.f. 9, $P_{D2\_SpeechWritten} < 0.001$, Cohen's $d = 2.34$; $P_{D2\_SpeechAuditory} < 0.001$, Cohen's $d = 3.23$). Representation for all words was observed in each phase, including pseudowords (bindip and nifzig) (Fig. 3c,f). To identify neurons with selective activity for unique words, we performed a Kruskal–Wallis test (Supplementary Fig. 3c,d). The results mirrored findings of the regression analysis in both participants, albeit weaker in participant 2. These findings suggest that, while neural activity during active phases differed from activity during the ITI phase, neural responses of only a few neurons varied across different words for participant 2.

The neural population in the SMG simultaneously represented several distinct aspects of language processing: temporal changes, input modality (auditory, written for participant 1) and unique words from our vocabulary list. We used demixed principal component analysis (dPCA) to decompose and analyse contributions of each individual component: timing, cue modality and word. In Fig. 4, demixed principal components (PCs) explaining the highest amount of variance were plotted by projecting data onto their respective dPCA decoder axis.

For participant 1, the 'timing' component revealed that temporal dynamics in the SMG peaked during all active phases (Fig. 4a). In contrast, temporal S1 modulation peaked only during vocalized speech production, indicating a lack of synchronized lip and face movement of the participant during the other task phases. While 'cue modality' components were separable during the cue phase (Fig. 4b), they overlapped during subsequent phases. Thus, internal and vocalized speech representation may not be influenced by the cue modality. Pseudowords had similar separability to lexical words (Fig. 4c). The explained variance between words was high in the SMG and was close to zero in S1. In participant 2, temporal dynamics of the task were preserved ('timing' component). However, variance to words was reduced, suggesting lower neuronal ability to represent individual words in participant 2. In S1, the results mirrored findings from S1 in participant 1 (Fig. 4d,e, right).

### Internal speech is decodable in the SMG

Separable neural representations of both internal and vocalized speech processes implicate SMG as a rich source of neural activity for real-time speech BMI devices. The decodability of words correlated with the percentage of tuned neurons (Fig. 3a–f) as well as the explained dPCA variance (Fig. 4c,e) observed in the participants. In participant 1, all words in our vocabulary list were highly decodable, averaging 55% offline decoding and 79% (16–20 training trials) online decoding from neurons during internal speech (Fig. 5a,b). Words spoken during the vocalized phase were also highly discriminable, averaging 74% offline (Fig. 5a). In participant 2, offline internal speech decoding averaged 24% (Supplementary Fig. 4b) and online decoding averaged 23% (Fig. 5a), with preferential representation of words 'spoon' and 'swimming'.

In participant 1, trial data from both types of cue (auditory and written) were concatenated for offline analysis, since SMG activity was only differentiable between the types of cue during the cue phase (Figs. 3a and 4b). This resulted in 16 trials per condition. Features were selected via principal component analysis (PCA) on the training dataset, and PCs that explained 95% of the variance were kept. A linear discriminant analysis (LDA) model was evaluated with leave-one-out cross-validation (CV). Significance was computed by comparing results with a null distribution (Methods).

Significant word decoding was observed during all phases, except during the ITI (Fig. 5a, $n = 10$, mean decoding value above 99.5th percentile of shuffle distribution is $P < 0.01$, per phase, Cohen's $d = 0.64$, 6.17, 3.04, 6.59, 3.93 and 8.26, confidence interval of the mean ± 1.73, 4.46, 5.21, 5.67, 4.63 and 6.49). Decoding accuracies were significantly higher in the cue, internal speech and speech condition, compared with rest phases ITI, D1 and D2 (Fig. 5a, paired $t$-test, $n = 10$, d.f. 9, for all $P < 0.001$, Cohen's $d = 6.81$, 2.29 and 5.75). Significant cue phase decoding suggested that modality-independent linguistic representations were present early within the task[45]. Internal speech decoding averaged 55% offline, with the highest session at 72% and a chance level of ~12.5% (Fig. 5a, red line). Vocalized speech averaged even higher, at 74%. All words were highly decodable (Fig. 5c). As suggested from our dPCA results, individual words were not significantly decodable from neural activity in S1 (Supplementary Fig. 4a), indicating generalized activity for vocalized speech in the S1 arm region (Fig. 4c).

For participant 2, SMG significant word decoding was observed during the cue, internal and vocalized speech phases (Supplementary Fig. 4b, $n = 9$, mean decoding value above 97.5th/99.5th percentile of shuffle distribution is $P < 0.05/P < 0.01$, per phase Cohen's $d = 0.35$, 1.15, 1.09, 1.44, 0.99 and 1.49, confidence interval of the mean ± 3.09, 5.02, 6.91, 8.14, 5.45 and 4.15). Decoding accuracies were significantly higher in the cue and internal speech condition, compared with rest phases ITI and D1 (Supplementary Fig. 4b, paired $t$-test, $n = 9$, d.f. 8, $P_{ITI\_Cue} = 0.013$, Cohen's $d = 1.07$, $P_{D1\_Internal} = 0.01$, Cohen's $d = 1.11$). S1 decoding mirrored results in participant 1, suggesting that no synchronized face movements occurred during the cue phase or internal speech phase (Supplementary Fig. 4c).

### High-accuracy online speech decoder

We developed an online, closed-loop internal speech BMI using an eight-word vocabulary (Fig. 5b). On three separate session days, training datasets were generated using the written cue task, with eight repetitions of each word for each participant. An LDA model was trained on the internal speech data of the training set, corresponding to only 1.5 s of neural data per repetition for each class. The trained decoder predicted internal speech during the online task. During the online task, the vocalized speech phase was replaced with a feedback phase. The decoded word was shown in green if correctly decoded, and in red if wrongly decoded (Supplementary Video 1). The classifier was retrained after each run of the online task, adding the newly recorded data. Several online runs were performed on each session day, corresponding to different datapoints on Fig. 5b. When using between 8 and

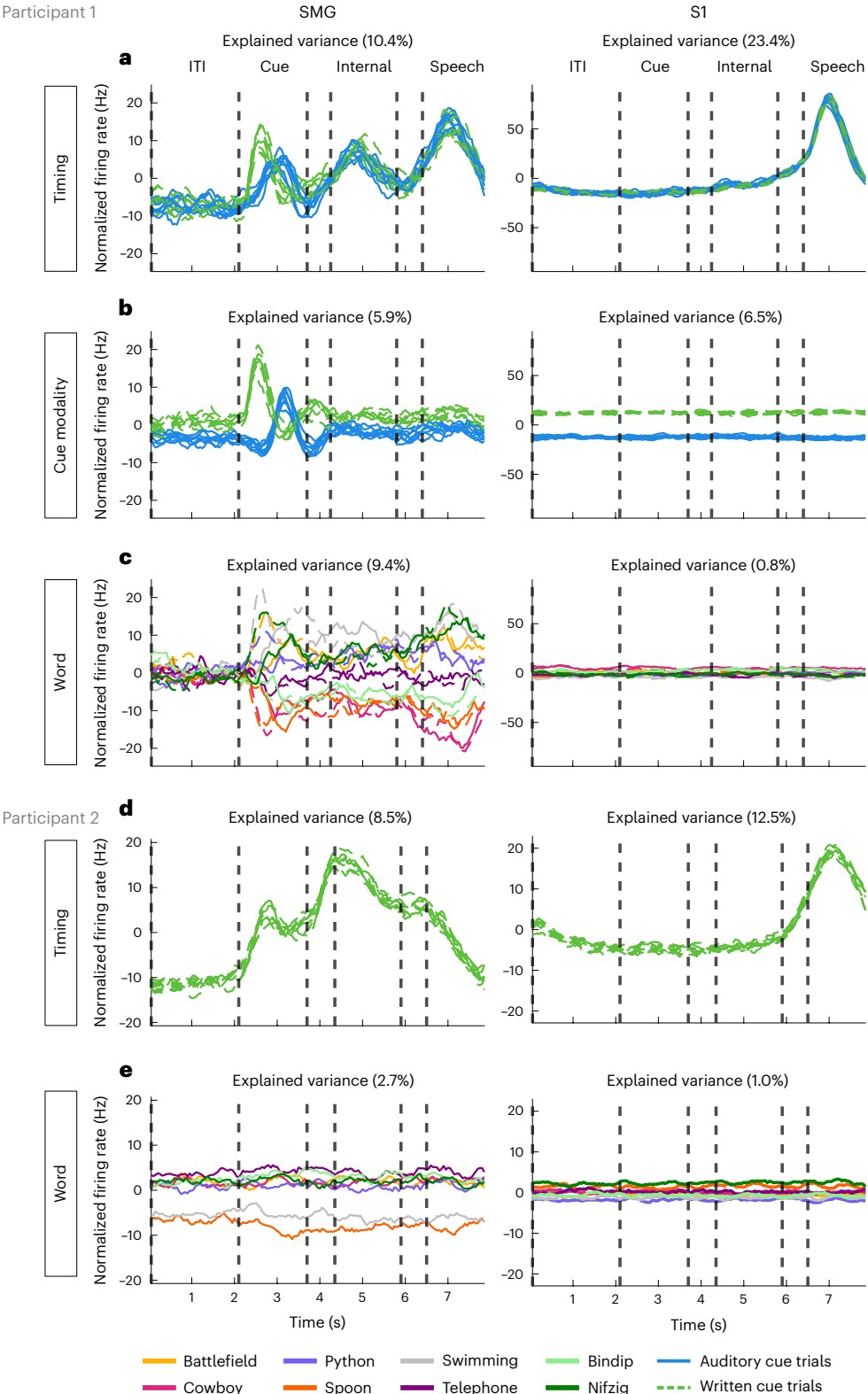

**Fig. 4 | dPCA highlighting SMG's involvement in language processing.**
**a**–**e**, dPCA was performed to investigate variance within three marginalizations: 'timing', 'cue modality' and 'word' for participant 1 (**a**–**c**) and 'timing' and 'word' for participant 2 (**d** and **e**). Demixed PCs explaining the highest variance within each marginalization were plotted over time, by projecting the data onto their respective dPCA decoder axis. In **a**, the 'timing' marginalization demonstrates SMG modulation during cue, internal speech and vocalized speech, while S1 only represents vocalized speech. The solid blue lines (8) represent the auditory cue trials, and dashed green lines (8) represent written cue trials. In **b**, the 'cue modality' marginalization suggests that internal and vocalized speech

representation in the SMG are not affected by the cue modality. The solid blue lines (8) represent the auditory cue trials, and dashed green lines (8) represent written cue trials. In **c**, the 'word' marginalization shows high variability for different words in the SMG, but near zero for S1. The colours (8) represent individual words. For each colour, solid lines represent auditory trials and dashed lines represent written cue trials. **d** is the same as **a**, but for participant 2. The dashed green lines (8) represent written cue trials. **e** is the same as **c**, but for participant 2. The colours (8) represent individual words during written cue trials. The variance for different words in the SMG (left) was higher than in S1 (right), but lower in comparison with SMG in participant 1 (**c**).

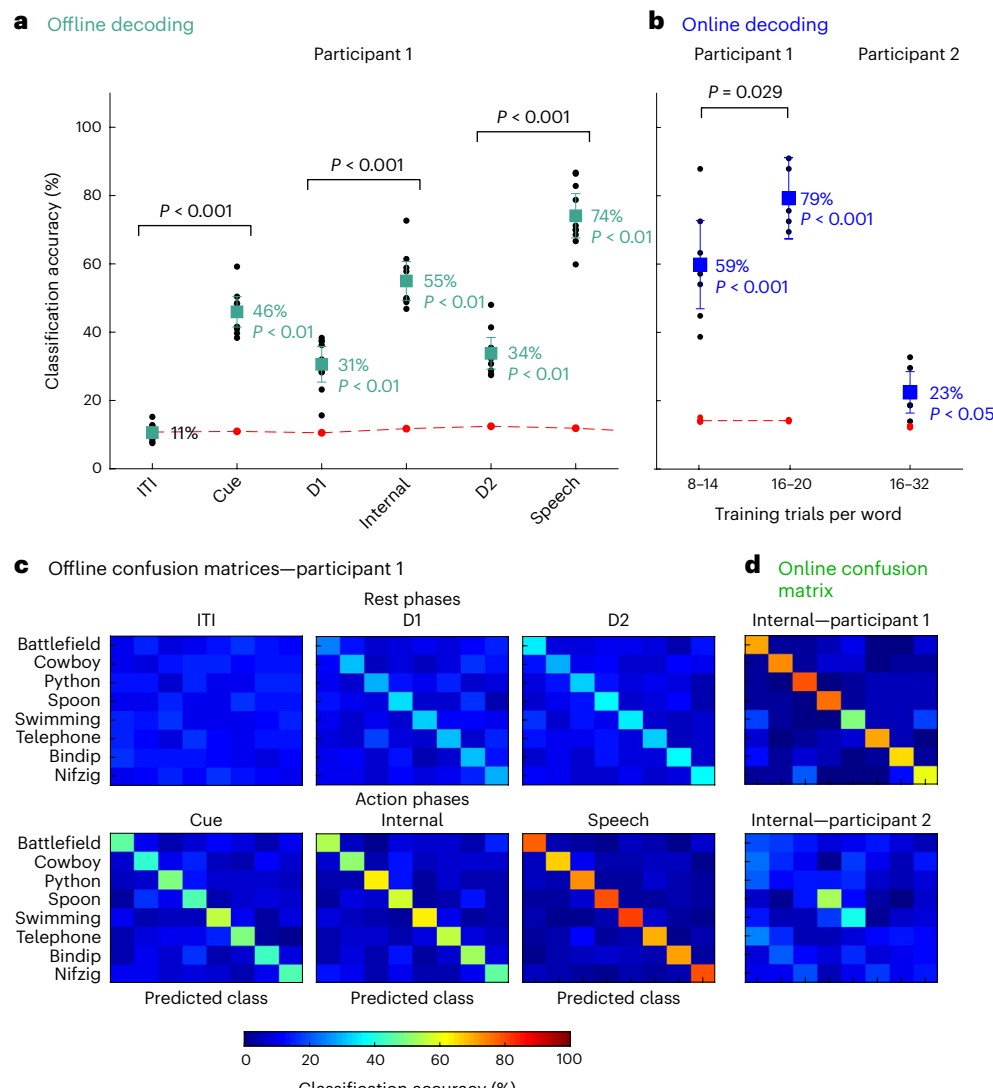

**Fig. 5 | Words can be significantly decoded during internal speech in the SMG. a**, Offline decoding accuracies: 'audio cue' and 'written cue' task data were combined for each individual session day, and leave-one-out CV was performed (black dots). PCA was performed on the training data, an LDA model was constructed, and classification accuracies were plotted with 95% confidence intervals, over the session means. The significance of classification accuracies were evaluated by comparing results with a shuffled distribution (averaged shuffle results over 100 repetitions indicated by red dots; $P < 0.01$ indicates that the average mean is >99.5th percentile of shuffle distribution, $n = 10$). In participant 1, classification accuracies during action phases (cue, internal and speech) following rest phases (ITI, D1 and D2) were significantly higher (paired two-tailed $t$-test: $n = 10$, d.f. 9, for all $P < 0.001$, Cohen's $d = 6.81, 2.29$ and $5.75$). **b**, Online decoding accuracies: classification accuracies for internal speech were evaluated in a closed-loop internal speech BMI application on three different session days for both participants. In participant 1, decoding accuracies were significantly above chance (averaged shuffle results over 1,000 repetitions indicated by red dots; $P < 0.001$ indicates that the average mean is >99.95th percentile of shuffle distribution) and improved when 16–20 trials per words were used to train the model (two-sample two-tailed $t$-test, $n_{(8-14)} = 8$, d.f. 11, $n_{(16-20)} = 5, P = 0.029$), averaging 79% classification accuracy. In participant 2, online decoding accuracies were significant (averaged shuffle results over 1,000 repetitions indicated by red dots; $P < 0.05$ indicates that average mean is >97.5th percentile of shuffle distribution, $n = 7$) and averaged 23%. **c**, An offline confusion matrix for participant 1: confusion matrices for each of the different task phases were computed on the tested data and averaged over all session days. **d**, An online confusion matrix: a confusion matrix was computed combining all online runs, leading to a total of 304 trials (38 trials per word) for participant 1 and 448 online trials for participant 2. Participant 1 displayed comparable online decoding accuracies for all words, while participant 2 had preferential decoding for the words 'swimming' and 'spoon'.

14 repetitions per words to train the decoding model, an average of 59% classification accuracy was obtained for participant 1. Accuracies were significantly higher (two-sample two-tailed $t$-test, $n_{(8-14)} = 8, n_{(16-20)} = 5$, d.f. 11, $P = 0.029$) the more data were added to train the model, obtaining an average of 79% classification accuracy with 16–20 repetitions per word. The highest single run accuracy was 91%. All words were well represented, illustrated by a confusion matrix of 304 trials (Fig. 5d). In participant 2, decoding was statistically significant, but lower compared with participant 1. The lower number of tuned units (Fig. 3a–f) and reduced explained variance between words (Fig. 4e, left) could

account for these findings. Additionally, preferential representation of words 'spoon' and 'swimming' was observed.

## Shared representations between internal speech, written words and vocalized speech

Different language processes are engaged during the task: auditory comprehension or visual word recognition during the cue phase, and internal speech and vocalized speech production during the speech phases. It has been widely assumed that each of these processes is part of a highly distributed network, involving multiple cortical areas[46].

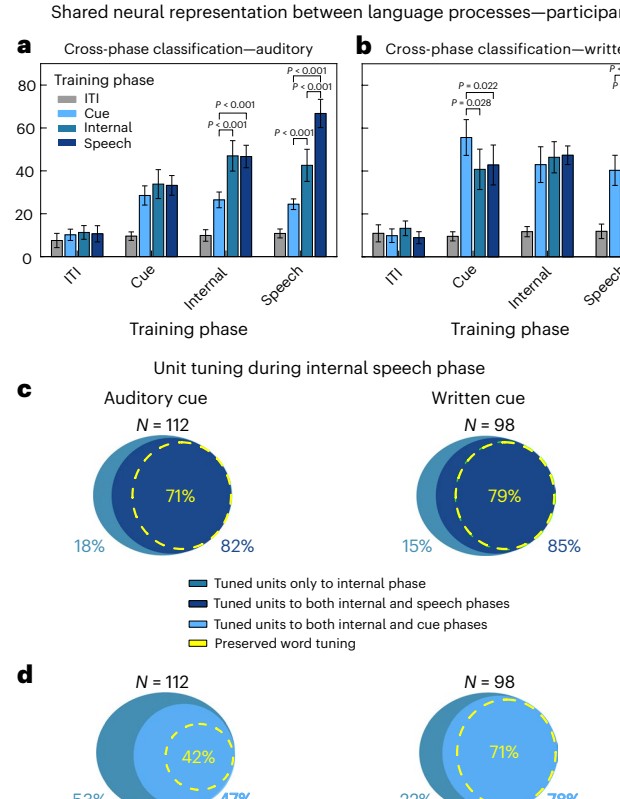

Shared neural representation between language processes—participant 1

**a** Cross-phase classification—auditory
**b** Cross-phase classification—written
**c** Unit tuning during internal speech phase

**Fig. 6 | Shared representations between internal speech, vocalized speech and written word processing. a**, Evaluating the overlap of shared information between different task phases in the 'auditory cue' task. For each of the ten session days, cross-phase classification was performed. It consisted in training a model on a subset of data from one phase (for example, cue) and applying it on a subset of data from ITI, cue, internal and speech phases. This analysis was performed separately for each task phase. PCA was performed on the training data, an LDA model was constructed and classification accuracies were plotted with a 95% confidence interval over session means. Significant differences in performance between phases were evaluated between the ten sessions (paired two-tailed $t$-test, FDR corrected, d.f. 9, $P < 0.001$ for all, Cohen's $d \geq 1.89$). For easier visibility, significant differences between ITI and other phases were not plotted. **b**, Same as **a** for the 'written cue' task (paired two-tailed $t$-test, FDR corrected, d.f. 9, $P_{Cue\_Internal} = 0.028$, Cohen's $d > 0.86$; $P_{Cue\_Speech} = 0.022$, Cohen's $d = 0.95$; all others $P < 0.001$ and Cohen's $d \geq 1.65$). **c**, The percentage of neurons tuned during the internal speech phase that are also tuned during the vocalized speech phase. Neurons tuned during the internal speech phase were computed as in Fig. 3b separately for each session day. From these, the percentage of neurons that were also tuned during vocalized speech was calculated. More than 80% of neurons during internal speech were also tuned during vocalized speech (82% in the 'auditory cue' task, 85% in the 'written cue' task). In total, 71% of 'auditory cue' and 79% 'written cue' neurons also preserved tuning to at least one identical word during internal speech and vocalized speech phases. **d**, The percentage of neurons tuned during the internal speech phase that were also tuned during the cue phase. Right: 78% of neurons tuned during internal speech were also tuned during the written cue phase. Left: a smaller 47% of neurons tuned during the internal speech phase were also tuned during the auditory cue phase. In total, 71% of neurons preserved tuning between the written cue phase and the internal speech phase, while 42% of neurons preserved tuning between the auditory cue and the internal speech phase.

In this work, we observed significant representation of different language processes in a common cortical region, SMG, in our participants. To explore the relationships between each of these processes, for participant 1 we used cross-phase classification to identify the distinct and common neural codes separately in the auditory and written cue datasets. By training our classifier on the representation found in one

phase (for example, the cue phase) and testing the classifier on another phase (for example, internal speech), we quantified generalizability of our models across neural activity of different language processes (Fig. 6). The generalizability of a model to different task phases was evaluated through paired $t$-tests. No significant difference between classification accuracies indicates good generalization of the model, while significantly lower classification accuracies suggest poor generalization of the model.

The strongest shared neural representations were found between visual word recognition, internal speech and vocalized speech (Fig. 6b). A model trained on internal speech was highly generalizable to both vocalized speech and written cued words, evidence for a possible shared neural code (Fig. 6b, internal). In contrast, the model's performance was significantly lower when tested on data recorded in the auditory cue phase (Fig. 6a, training phase internal: paired $t$-test, d.f. 9, $P_{Cue\_Internal} < 0.001$, Cohen's $d = 2.16$; $P_{Cue\_Speech} < 0.001$, Cohen's $d = 3.34$). These differences could stem from the inherent challenges in comparing visual and auditory language stimuli, which differ in processing time: instantaneous for text versus several hundred milliseconds for auditory stimuli.

We evaluated the capability of a classification model, initially trained to distinguish words during vocalized speech, in its ability to generalize to internal and cue phases (Fig. 6a,b, training phase speech). The model demonstrated similar levels of generalization during internal speech and in response to written cues, as indicated by the lack of significance in decoding accuracy between the internal and written cue phase (Fig. 6b, training phase speech, cue–internal). However, the model generalized significantly better to internal speech than to representations observed during the auditory cue phase (Fig. 6a, training phase speech, d.f. 9, $P_{Cue\_Internal} < 0.001$, Cohen's $d = 2.85$).

Neuronal representation of words at the single-neuron level was highly consistent between internal speech, vocalized speech and written cue phases. A high percentage of neurons were not only active during the same task phases but also preserved identical tuning to at least one word (Fig. 6c,d). In total, 82–85% of neurons active during internal speech were also active during vocalized speech. In 71–79% of neurons, tuning was preserved between the internal speech and vocalized speech phases (Fig. 6c). During the cue phase, 78% of neurons active during internal speech were also active during the written cue (Fig. 6d, right). However, a lower percentage of neurons (47%) were active during the auditory cue phase (Fig. 6d, left). Similarly, 71% of neurons preserved tuning between the written cue phase and the internal speech phase, while 42% of neurons preserved tuning between the auditory cue phase and the internal speech phase.

Together with the cross-phase analysis, these results suggest strong shared neural representations between internal speech, vocalized speech and the written cue, both at the single-neuron and at the population level.

## Robust decoding of multiple internal speech strategies within the SMG

Strong shared neural representations in participant 1 between written, inner and vocalized speech suggest that all three partly represent the same cognitive process or all cognitive processes share common neural features. While internal and vocalized speech have been shown to share common neural features[36], similarities between internal speech and the written cue could have occurred through several different cognitive processes. For instance, the participant's observation of the written cue could have activated silent reading. This process has been self-reported as activating internal speech, which can involve 'hearing' a voice, thus having an auditory component[42,47]. However, the participant could also have mentally pictured an image of the written word while performing internal speech, involving visual imagination in addition to language processes. Both hypotheses could explain the high amount of shared neural representation between the written cue and the internal speech phases (Fig. 6b).

**Table 1 | Internal strategy task**

| Task variation | Cue modality | Internal strategy |
|---|---|---|
| Auditory–sound | Auditory | Sound imagination |
| Written–visual | Written | Visual imagination |
| Auditory–visual | Auditory | Visual imagination |
| Written–sound | Written | Sound imagination |

We therefore compared two possible internal sensory strategies in participant 1: a 'sound imagination' strategy in which the participant imagined hearing the word, and a 'visual imagination' strategy in which the participant visualized the word's image (Supplementary Fig. 5a). Each strategy was cued by the modalities we had previously tested (auditory and written words) (Table 1). To assess the similarity of these internal speech processes to other task phases, we conducted a cross-phase decoding analysis (as performed in Fig. 6). We hypothesized that, if the high cross-decoding results between internal and written cue phases primarily stemmed from the participant engaging in visual word imagination, we would observe lower decoding accuracies during the auditory imagination phase.

Both strategies demonstrated high representation of the four-word dataset (Supplementary Fig. 5b, highest 94%, chance level 25%). These results suggest our speech BMI decoder is robust to multiple types of internal speech strategy.

The participant described the 'sound imagination' strategy as being easier and more similar to the internal speech condition of the first experiment. The participant's self-reported strategy suggests that no visual imagination was performed during internal speech. Correspondingly, similarities between written cue and internal speech phases may stem from internal speech activation during the silent reading of the cue.

## Discussion

In this work, we demonstrated a decoder for internal and vocalized speech, using single-neuron activity from the SMG. Two chronically implanted, speech-abled participants with tetraplegia were able to use an online, closed-loop internal speech BMI to achieve on average 79% and 23% classification accuracy with 16–32 training trials for an eight-word vocabulary. Furthermore, high decoding was achievable with only 24 s of training data per word, corresponding to 16 trials each with 1.5 s of data. Firing rates recorded from S1 showed generalized activation only during vocalized speech activity, but individual words were not classifiable. In the SMG, shared neural representations between internal speech, the written cue and vocalized speech suggest the occurrence of common processes. Robust control could be achieved using visual and auditory internal speech strategies. Representation of pseudowords provided evidence for a phonetic word encoding component in the SMG.

### Single neurons in the SMG encode internal speech

We demonstrated internal speech decoding of six different words and two pseudowords in the SMG. Single neurons increased their firing rates during internal speech (Fig. 2, S1 and S2), which was also reflected at the population level (Fig. 3a,b,d,e). Each word was represented in the neuronal population (Fig. 3c,f). Classification accuracy and tuning during the internal speech phase were significantly higher than during the previous delay phase (Figs. 3b,e and 5a, and Supplementary Figs. 3c,d and 4b). This evidence suggests that we did not simply decode sustained activity from the cue phase but activity generated by the participant performing internal speech. We obtained significant offline and online internal speech decoding results in two participants (Fig. 5a and Supplementary Fig. 4b). These findings provide strong evidence for internal speech processing at the single-neuron level in the SMG.

### Neurons in S1 are modulated by vocalized but not internal speech

Neural activity recorded from S1 served as a control for synchronized face and lip movements during internal speech. While vocalized speech robustly activated sensory neurons, no increase of baseline activity was observed during the internal speech phase or the auditory and written cue phases in both participants (Fig. 4, S1). These results underline no synchronized movement inflated our decoding accuracy of internal speech (Supplementary Fig. 4a,c).

A previous imaging study achieved significant offline decoding of several different internal speech sentences performed by patients with mild ALS[6]. Together with our findings, these results suggest that a BMI speech decoder that does not rely on any movement may translate to communication opportunities for patients suffering from ALS and locked-in syndrome.

### Different face activities are observable but not decodable in arm area of S1

The topographic representation of body parts in S1 has recently been found to be less rigid than previously thought. Generalized finger representation was found in a presumably S1 arm region of interest (ROI)[44]. Furthermore, an fMRI paper found observable face and lip activity in S1 leg and hand ROIs. However, differentiation between two lip actions was restricted to the face ROI[43]. Correspondingly, we observed generalized face and lip activity in a predominantly S1 arm region for participant 1 (see ref. 48 for implant location) and a predominantly S1 hand region for participant 2 during vocalized speech (Fig. 4a,d and Supplementary Figs. 1 and 4a,b). Recorded neural activity contained similar representations for different spoke words (Fig. 4c,e) and was not significantly decodable (Supplementary Fig. 4a,c).

### Shared neural representations between internal and vocalized speech

The extent to which internal and vocalized speech generalize is still debated[35,42,49] and depends on the investigated brain area[36,50]. In this work, we found on average stronger representation for vocalized (74%) than internal speech (Fig. 5a, 55%) in participant 1 but the opposite effect in participant 2 (Supplementary Fig. 4b, 24% internal, 21% vocalized speech). Additionally, cross-phase decoding of vocalized speech from models trained on data during internal speech resulted in comparable classification accuracies to those of internal speech (Fig. 6a,b, internal). Most neurons tuned during internal speech were also tuned to at least one of the same words during vocalized speech (71–79%; Fig. 6c). However, some neurons were only tuned during internal speech, or to different words. These observations also applied to firing rates of individual neurons. Here, we observed neurons that had higher peak rates during the internal speech phase than the vocalized speech phase (Supplementary Fig. 1: swimming and cowboy). Together, these results further suggest neural signatures during internal and vocalized speech are similar but distinct from one another, emphasizing the need for developing speech models from data recorded directly on internal speech production[51].

Similar observations were made when comparing internal speech processes with visual word processes. In total, 79% of neurons were active both in the internal speech phase and the written cue phase, and 79% preserved the same tuning (Fig. 6d, written cue). Additionally, high cross-decoding between both phases was observed (Fig. 6b, internal).

### Shared representation between speech and written cue presentation

Observation of a written cue may engage a variety of cognitive processes, such as visual feature recognition, semantic understanding and/or related language processes, many of which modulate similar cortical regions as speech[45]. Studies have found that silent reading can evoke internal speech; it can be modulated by a presumed author's

speaking speed, voice familiarity or regional accents[35,42,47,52,53]. During silent reading of a cued sentence with a neutral versus increased prosody (madeleine brought me versus MADELEINE brought me), one study in particular found that increased left SMG activation correlated with the intensity of the produced inner speech[54].

Our data demonstrated high cross-phase decoding accuracies between both written cue and speech phases in our first participant (Fig. 6b). Due to substantial shared neural representation, we hypothesize that the participant's silent reading during the presentation of the written cue may have engaged internal speech processes. However, this same shared representation could have occurred if visual processes were activated in the internal speech phase. For instance, the participant could have performed mental visualization of the written word instead of generating an internal monologue, as the subjective perception of internal speech may vary between individuals.

### Investigating internal speech strategies

In a separate experiment, participant 1 was prompted to execute different mental strategies during the internal speech phase, consisting of 'sound imagination' or 'visual word imagination' (Supplementary Fig. 5a). We found robust decoding during the internal strategy phase, regardless of which mental strategy was performed (Supplementary Fig. 5b). This participant reported the sound strategy was easier to execute than the visual strategy. Furthermore, this participant reported that the sound strategy was more similar to the internal speech strategy employed in prior experiments. This self-report suggests that the patient did not perform visual imagination during the internal speech task. Therefore, shared neural representation between internal and written word phases during the internal speech task may stem from silent reading of the written cue. Since multiple internal mental strategies are decodable from SMG, future patients could have flexibility with their preferred strategy. For instance, people with a strong visual imagination may prefer performing visual word imagination.

### Audio contamination in decoding result

Prior studies examining neural representation of attempted or vocalized speech must potentially mitigate acoustic contamination of electrophysiological brain signals during speech production[55]. During internal speech production, no detectable audio was captured by the audio equipment or noticed by the researchers in the room. In the rare cases the participant spoke during internal speech (three trials), the trials were removed. Furthermore, if audio had contaminated the neural data during the auditory cue or vocalized speech, we would have probably observed significant decoding in all channels. However, no significant classification was detected in S1 channels during the auditory cue phase nor the vocalized speech phase (Supplementary Fig. 2b). We therefore conclude that acoustic contamination did not artificially inflate observed classification accuracies during vocalized speech in the SMG.

### Single-neuron modulation during internal speech with a second participant

We found single-neuron modulation to speech processes in a second participant (Figs. 2d,e and 3f, and Supplementary Fig. 2d), as well as significant offline and online classification accuracies (Fig. 5a and Supplementary Fig. 4b), confirming neural representation of language processes in the SMG. The number of neurons distinctly active for different words was lower compared with the first participant (Fig. 2e and Supplementary Fig. 3d), limiting our ability to decode with high accuracy between words in the different task phases (Fig. 5a and Supplementary Fig. 4b).

Previous work found that single neurons in the PPC exhibited a common neural substrate for written action verbs and observed actions[56]. Another study found that single neurons in the PPC also encoded spoken numbers[57]. These recordings were made in the

superior parietal lobule whereas the SMG is in the inferior parietal lobule. Thus, it would appear that language-related activity is highly distributed across the PPC. However, the difference in strength of language representation between each participant in the SMG suggests that there is a degree of functional segregation within the SMG[37].

Different anatomical geometries of the SMG between participants mean that precise comparisons of implanted array locations become difficult (Fig. 1). Implant locations for both participants were informed from pre-surgical anatomical/vasculature scans and fMRI tasks designed to evoke activity related to grasp and dexterous hand movements[48]. Furthermore, the number of electrodes of the implanted array was higher in the first participant (96) than in the second participant (64). A pre-surgical assessment of functional activity related to language and speech may be required to determine the best candidate implant locations within the SMG for online speech decoding applications.

### Impact on BMI applications

In this work, an online internal speech BMI achieved significant decoding from single-neuron activity in the SMG in two participants with tetraplegia. The online decoders were trained on as few as eight repetitions of 1.5 s per word, demonstrating that meaningful classification accuracies can be obtained with only a few minutes' worth of training data per day. This proof-of-concept suggests that the SMG may be able to represent a much larger internal vocabulary. By building models on internal speech directly, our results may translate to people who cannot vocalize speech or are completely locked in. Recently, ref. 26 demonstrated a BMI speller that decoded attempted speech of the letters of the NATO alphabet and used those to construct sentences. Scaling our vocabulary to that size could allow for an unrestricted internal speech speller.

To summarize, we demonstrate the SMG as a promising candidate to build an internal brain–machine speech device. Different internal speech strategies were decodable from the SMG, allowing patients to use the methods and languages with which they are most comfortable. We found evidence for a phonetic component during internal and vocalized speech. Adding to previous findings indicating grasp decoding in the SMG[23], we propose the SMG as a multipurpose BMI area.

## Methods

### Experimental model and participant details

Two male participants with tetraplegia (33 and 39 years) were recruited for an institutional review board- and Food and Drug Administration-approved clinical trial of a BMI and gave informed consent to participate (Institutional Review Board of Rancho Los Amigos National Rehabilitation Center, Institutional Review Board of California Institute of Technology, clinical trial registration NCT01964261). This clinical trial evaluated BMIs in the PPC and the somatosensory cortex for grasp rehabilitation. One of the primary effectiveness objectives of the study is to evaluate the effectiveness of the neuroport in controlling virtual or physical end effectors. Signals from the PPC will allow the subjects to control the end effector with accuracy greater than chance. Participants were compensated for their participation in the study and reimbursed for any travel expenses related to participation in study activities. The authors affirm that the human research participant provided written informed consent for publication of Supplementary Video 1. The first participant suffered a spinal cord injury at cervical level C5 1.5 years before participating in the study. The second participant suffered a C5–C6 spinal cord injury 3 years before implantation.

### Method details

**Implants.** Data were collected from implants located in the left SMG and the left S1 (for anatomical locations, see Fig. 1). For description of pre-surgical planning, localization fMRI tasks, surgical techniques and methodologies, see ref. 48. Placement of electrodes was based on fMRI tasks involving grasp and dexterous hand movements.

The first participant underwent surgery in November 2016 to implant two 96-channel platinum-tipped multi-electrode arrays (NeuroPort Array, Blackrock Microsystems) in the SMG and in the ventral premotor cortex and two 7 × 7 sputtered iridium oxide film (SIROF)-tipped microelectrode arrays with 48 channels each in the hand and arm area of S1. Data were collected between July 2021 and August 2022. The second participant underwent surgery in October 2022 and was implanted with SIROF-tipped 64-channel microelectrode arrays in S1 (two arrays), SMG, ventral premotor cortex and primary motor cortex. Data were collected in January 2023.

**Data collection.** Recording began 2 weeks after surgery and continued one to three times per week. Data for this work were collected between 2021 and 2023. Broadband electrical activity was recorded from the NeuroPort Arrays using Neural Signal Processors (Blackrock Microsystems). Analogue signals were amplified, bandpass filtered (0.3–7,500 Hz) and digitized at 30,000 samples s⁻¹. To identify putative action potentials, these broadband data were bandpass filtered (250–5,000 Hz) and thresholded at −4.5 the estimated root-mean-square voltage of the noise. For some of the analyses, waveforms captured at these threshold crossings were then spike sorted by manually assigning each observation to a putative single neuron; for others, multiunit activity was considered. For participant 1, an average of 33 sorted SMG units (between 22 and 56) and 83 sorted S1 units (between 59 and 96) were recorded per session. For participant 2, an average of 80 sorted SMG units (between 69 and 92) and 81 sorted S1 units (between 61 and 101) were recorded per session. Auditory data were recorded at 30,000 Hz simultaneously to the neural data. Background noise was reduced post-recording by using the noise reduction function of the program 'Audible'.

## Experimental tasks
We implemented different tasks to study language processes in the SMG. The tasks cued six words informed by ref. 31 (spoon, python, battlefield, cowboy, swimming and telephone) as well as two pseudowords (bindip and nifzig). The participants were situated 1 m in front of a light-emitting diode screen (1,190 mm screen diagonal), where the task was visualized. The task was implemented using the Psychophysics Toolbox[58–60] extension for MATLAB. Only the written cue task was used for participant 2.

**Auditory cue task.** Each trial consisted of six phases, referred to in this paper as ITI, cue, D1, internal, D2 and speech. The trial began with a brief ITI (2 s), followed by a 1.5-s-long cue phase. During the cue phase, a speaker emitted the sound of one of the eight words (for example, python). Word duration varied between 842 and 1,130 ms. Then, after a delay period (grey circle on screen; 0.5 s), the participant was instructed to internally say the cued word (orange circle on screen; 1.5 s). After a second delay (grey circle on screen; 0.5 s), the participant vocalized the word (green circle on screen, 1.5 s).

**Written cue task.** The task was identical to the auditory cue task, except words were cued in writing instead of sound. The written word appeared on the screen for 1.5 s during the cue phase. The auditory cue was played between 200 ms and 650 ms later than the written cue appeared on the screen, due to the utilization of varied sound outputs (direct computer audio versus Bluetooth speaker).

One auditory cue task and one written cue task were recorded on ten individual session days in participant 1. The written cue task was recorded on seven individual session days in participant 2.

**Control experiments.** Three experiments were run to investigate internal strategies and phonetic versus semantic processing.

**Internal strategy task.** The task was designed to vary the internal strategy employed by the participant during the internal speech phase.

Two internal strategies were tested: a sound imagination and a visual imagination. For the 'sound imagination' strategy, the participant was instructed to imagine what the sound of the word sounded like. For the 'visual imagination' strategy, the participant was instructed to perform mental visualization from the written word. We also tested if the cue modality (auditory or written) influenced the internal strategy. A subset of four words were used for this experiment. This led to four different variations of the task.

The internal strategy task was run on one session day with participant 1.

**Online task.** The 'written cue task' was used for the closed-loop experiments. To obtain training data for the online task, a written cue task was run. Then, a classification model was trained only on the internal speech data of the task (see 'Classification' section). The closed-loop task was nearly identical to the 'written cue task' but replaced the vocalized speech phase by a feedback phase. Feedback was provided by showing the word on the screen either in green if correctly classified or in red if wrongly classified. See Supplementary Video 1 for an example of the participant performing the online task. The online task was run on three individual session days.

**Error trials.** Trials in which participants accidentally spoke during the internal speech part (3 trials) or said the wrong word during the vocalized speech part (20 trials) were removed from all analysis.

**Total number of recording trials.** For participant 1, we collected offline datasets composed of eight trials per word across ten sessions. Trials during which participant errors occurred were excluded. In total, between 156 and 159 trials per word were included, with a total of 1,257 trials for offline analysis. On four non-consecutive session days, the auditory cue task was run first, and on six non-consecutive days, the written cue task was run first. For online analysis, datasets were recorded on three different session days, for a total of 304 trials. Participant 2 underwent a similar data collection process, with offline datasets comprising 16 trials per word using the written cue modality over nine sessions. Error trials were excluded. In total, between 142 and 144 trials per word were kept, with a total of 1,145 trials for offline analysis. For online analysis, datasets were recorded on three session days, leading to a total of 448 online trials.

## Quantification and statistical analysis
Analyses were performed using MATLAB R2020b and Python, version 3.8.11.

**Neural firing rates.** Firing rates of sorted units were computed as the number of spikes occurring in 50-ms bins, divided by the bin width and smoothed using a Gaussian filter with kernel width of 50 ms to form an estimate of the instantaneous firing rates (spikes s⁻¹).

**Linear regression tuning analysis.** To identify units exhibiting selective firing rate patterns (or tuning) for each of the eight words, linear regression analysis was performed in two different ways: (1) step by step in 50-ms time bins to allow assessing changes in neuronal tuning over the entire trial duration; (2) averaging the firing rate in each task phase to compare tuning between phases. The model returns a fit that estimates the firing rate of a unit on the basis of the following variables:

$$FR = \sum_{w=1}^{W} \beta_w X_w + \beta_0,$$

where FR corresponds to the firing rate of the unit, $\beta_0$ to the offset term equal to the average ITI firing rate of the unit, **X** is the vector indicator variable for each word $w$, and $\beta_w$ corresponds to the estimated regression coefficient for word $w$. $W$ was equal to 8 (battlefield, cowboy, python, spoon, swimming, telephone, bindip and nifzig)[23].

In this model, $\beta$ symbolizes the change of firing rate from baseline for each word. A $t$-statistic was calculated by dividing each $\beta$ coefficient by its standard error. Tuning was based on the $P$ value of the $t$-statistic for each $\beta$ coefficient. A follow-up analysis was performed to adjust for false discovery rate (FDR) between the $P$ values[61,62]. A unit was defined as tuned if the adjusted $P$ value is <0.05 for at least one word. This definition allowed for tuning of a unit to zero, one or multiple words during different timepoints of the trial. Linear regression was performed for each session day individually. A 95% confidence interval of the mean was computed by performing the Student's $t$-inverse cumulative distribution function over the ten sessions.

**Kruskal–Wallis tuning analysis.** As an alternative tuning definition, differences in firing rates between words were tested using the Kruskal–Wallis test, the non-parametric analogue to the one-way analysis of variance (ANOVA). For each neuron, the analysis was performed to evaluate the null hypothesis that data from each word come from the same distribution. A follow-up analysis was performed to adjust for FDR between the $P$ values for each task phase[61,62]. A unit was defined as tuned during a phase if the adjusted $P$ value was smaller than $\alpha = 0.05$.

**Classification.** Using the neuronal firing rates recorded during the tasks, a classifier was used to evaluate how well the set of words could be differentiated during each phase. Classifiers were trained using averaged firing rates over each task phase, resulting in six matrices of size $n, m$, where $n$ corresponds to the number of trials and $m$ corresponds to the number of recorded units. A model for each phase was built using LDA, assuming an identical covariance matrix for each word, which resulted in best classification accuracies. Leave-one-out CV was performed to estimate decoding performance, leaving out a different trial across neurons at each loop. PCA was applied on the training data, and PCs explaining more than 95% of the variance were selected as features and applied to the single testing trial. A 95% confidence interval of the mean was computed as described above.

**Cross-phase classification.** To estimate shared neural representations between different task phases, we performed cross-phase classification. The process consisted in training a classification model (as described above) on one of the task phases (for example, ITI) and to test it on the ITI, cue, imagined speech and vocalized speech phases. The method was repeated for each of the ten sessions individually, and a 95% confidence interval of the mean was computed. Significant differences in classification accuracies between phases decoded with the same model were evaluated using a paired two-tailed $t$-test. FDR correction of the $P$ values was performed ('Linear regression tuning analysis')[61,62].

**Classification performance significance testing.** To assess the significance of classification performance, a null dataset was created by repeating classification 100 times with shuffled labels. Then, different percentile levels of this null distribution were computed and compared to the mean of the actual data. Mean classification performances higher than the 97.5th percentile were denoted with $P < 0.05$ and higher than 99.5th percentile were denoted with $P < 0.01$.

**dPCA analysis.** dPCA was performed on the session data to study the activity of the neuronal population in relation to the external task parameters: cue modality and word. Kobak et al.[63] introduced dPCA as a refinement of their earlier dimensionality reduction technique (of the same name) that attempts to combine the explanatory strengths of LDA and PCA. By deconstructing neuronal population activity into individual components, each component relates to a single task parameter[64].

This text follows the methodology outlined by Kobak et al.[63]. Briefly, this involved the following steps for $N$ neurons:

First, unlike in PCA, we focused not on the matrix, $X$, of the original data, but on the matrices of marginalizations, $X_\phi$. The marginalizations

were computed as neural activity averaged over trials, $k$, and some task parameters in analogy to the covariance decomposition done in multivariate analysis of variance. Since our dataset has three parameters: timing, $t$, cue modality, $c$ (for example, auditory or visual), and word, $w$ (eight different words), we obtained the total activity as the sum of the average activity with the marginalizations and a final noise term

$$X_{tcwk} = \bar{X} + \bar{X}_t + \underbrace{\bar{X}_c + \bar{X}_{tc}}_{\bar{X}_{tc}} + \underbrace{\bar{X}_w + \bar{X}_{tw}}_{\bar{X}_{tw}} + \underbrace{\bar{X}_{cw} + \bar{X}_{tcw}}_{\bar{X}_{tcw}} + \epsilon_{tcwk}.$$

The above notation of Kobak et al. is the same as used in factorial ANOVA, that is, $X_{tcwk}$ is the matrix of firing rates for all neurons, $< \cdot >_{ab}$ is the average over a set of parameters $a, b, \ldots$, $\bar{X} = <X_{tcwk}>_{tcwk}$, $\bar{X}_t = <X_{tcwk} - \bar{X}>_{cwk}$, $\bar{X}_{tc} = <X_{tcwk} - \bar{X} - \bar{X}_t - \bar{X}_c - \bar{X}_w>_{wk}$ and so on. Finally, $\epsilon_{tcwk} = X_{tcwk} - <X_{tcwk}>_k$.

Participant 1 datasets were composed of $N = 333$ (SMG), $N = 828$ (S1) and $k = 8$. Participant 2 datasets were composed of $N = 547$ (SMG), $N = 522$ (S1) and $k = 16$. To create balanced datasets, error trials were replaced by the average firing rate of $k - 1$ trials.

Our second step reduced the number of terms by grouping them as seen by the braces in the equation above, since there is no benefit in demixing a time-independent pure task, $a$, term $\bar{X}_a$ from the time–task interaction terms $\bar{X}_{ta}$ since all components are expected to change with time. The above grouping reduced the parametrization down to just five marginalization terms and the noise term (reading in order): the mean firing rate, the task-independent term, the cue modality term, the word term, the cue modality–word interaction term and the trial-to-trial noise.

Finally, we gained extra flexibility by having two separate linear mappings $F_\phi$ for encoding and $D_\phi$ for decoding (unlike in PCA, they are not assumed to be transposes of each other). These matrices were chosen to minimize the loss function (with a quadratic penalty added to avoid overfitting):

$$L_\phi = \|X_\phi - F_\phi D_\phi X\|^2 + \mu \|F_\phi D_\phi\|^2$$

Here, $\mu = (\lambda \|X\|)^2$, where $\lambda$ was optimally selected through tenfold CV in each dataset.

We refer the reader to Kobak et al. for a description of the full analytic solution.

**Reporting summary**
Further information on research design is available in the Nature Portfolio Reporting Summary linked to this article.

## Data availability
The data supporting the findings of this study are openly available via Zenodo at https://doi.org/10.5281/zenodo.10697024 (ref. 65). Source data are provided with this paper.

## Code availability
The custom code developed for this study is openly available via Zenodo at https://doi.org/10.5281/zenodo.10697024 (ref. 65).

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

## Acknowledgements
We thank L. Bashford and I. Rosenthal for helpful discussions and data collection. We thank our study participants for their dedication to the study that made this work possible. This research was supported by the NIH National Institute of Neurological Disorders and Stroke Grant U01: U01NS098975 and U01: U01NS123127 (S.K.W., D.A.B., K.P., C.L. and R.A.A.) and by the T&C Chen Brain-Machine Interface Center (S.K.W., D.A.B. and R.A.A.). The funders had no role in study design, data collection and analysis, decision to publish or preparation of the paper.

## Author contributions
S.K.W., D.A.B. and R.A.A. designed the study. S.K.W. and D.A.B. developed the experimental tasks and collected the data. S.K.W. analysed the results and generated the figures. S.K.W., D.A.B. and R.A.A. interpreted the results and wrote the paper. K.P. coordinated regulatory requirements of clinical trials. C.L. and B.L. performed the surgery to implant the recording arrays.

## Competing interests
The authors declare no competing interests.

## Additional information

**Correspondence and requests for materials** should be addressed to Sarah K. Wandelt.

# Reporting Summary

## Statistics

For all statistical analyses, confirm that the following items are present in the figure legend, table legend, main text, or Methods section.

| n/a | Confirmed | |
|---|---|---|
| ☐ | ☒ | The exact sample size (*n*) for each experimental group/condition, given as a discrete number and unit of measurement |
| ☐ | ☒ | A statement on whether measurements were taken from distinct samples or whether the same sample was measured repeatedly |
| ☐ | ☒ | The statistical test(s) used AND whether they are one- or two-sided *Only common tests should be described solely by name; describe more complex techniques in the Methods section.* |
| ☐ | ☒ | A description of all covariates tested |
| ☐ | ☒ | A description of any assumptions or corrections, such as tests of normality and adjustment for multiple comparisons |
| ☐ | ☒ | A full description of the statistical parameters including central tendency (e.g. means) or other basic estimates (e.g. regression coefficient) AND variation (e.g. standard deviation) or associated estimates of uncertainty (e.g. confidence intervals) |
| ☐ | ☒ | For null hypothesis testing, the test statistic (e.g. *F*, *t*, *r*) with confidence intervals, effect sizes, degrees of freedom and *P* value noted *Give P values as exact values whenever suitable.* |
| ☒ | ☐ | For Bayesian analysis, information on the choice of priors and Markov chain Monte Carlo settings |
| ☒ | ☐ | For hierarchical and complex designs, identification of the appropriate level for tests and full reporting of outcomes |
| ☒ | ☐ | Estimates of effect sizes (e.g. Cohen's *d*, Pearson's *r*), indicating how they were calculated |

*Our web collection on statistics for biologists contains articles on many of the points above.*

## Software and code

Policy information about availability of computer code

| Data collection | Neuroport System Blackrock Microsystems |
|---|---|
| Data analysis | Matlab 2022a,b. Python 3.10.8, Psychophysics Toolbox extension for MATLAB (2018) |

For manuscripts utilizing custom algorithms or software that are central to the research but not yet described in published literature, software must be made available to editors and reviewers. We strongly encourage code deposition in a community repository (e.g. GitHub). See the Nature Portfolio guidelines for submitting code & software for further information.

## Data

Policy information about availability of data

All manuscripts must include a data availability statement. This statement should provide the following information, where applicable:

- Accession codes, unique identifiers, or web links for publicly available datasets
- A description of any restrictions on data availability
- For clinical datasets or third party data, please ensure that the statement adheres to our policy

The custom code developed for this study is openly available at https://doi.org/10.5281/zenodo.10697024. The data supporting the findings of this study are openly available at https://doi.org/10.5281/zenodo.10697024.

# Research involving human participants, their data, or biological material

Policy information about studies with <u>human participants or human data</u>. See also policy information about <u>sex, gender (identity/presentation), and sexual orientation</u> and <u>race, ethnicity and racism</u>.

| | |
|---|---|
| Reporting on sex and gender | These data were not reported for privacy concerns for the participant. |
| Reporting on race, ethnicity, or other socially relevant groupings | These data were not reported for privacy concerns for the participant. |
| Population characteristics | These data were not reported for privacy concerns for the participant. |
| Recruitment | Participants were recruited through our collaborating hospitals according to IRB approved inclusion and exclusion criterion. |
| Ethics oversight | Institutional Review Board of Rancho Los Amigos National Rehabilitation Center, Institutional Review Board of California Institute of Technology |

Note that full information on the approval of the study protocol must also be provided in the manuscript.

# Field-specific reporting

Please select the one below that is the best fit for your research. If you are not sure, read the appropriate sections before making your selection.

☐ Life sciences    ☒ Behavioural & social sciences    ☐ Ecological, evolutionary & environmental sciences

For a reference copy of the document with all sections, see <u>nature.com/documents/nr-reporting-summary-flat.pdf</u>

# Behavioural & social sciences study design

All studies must disclose on these points even when the disclosure is negative.

| | |
|---|---|
| Study description | Data were quantitative experimental. Neuronal firing rates were recorded while a participant performed an internal and vocalized speech task. Recorded data were both analysed offline and in real time. |
| Research sample | Two human participants affected by tetraplegia performed internal and vocalized speech tasks. For participant 1, an average of 33 sorted SMG units (between 22–56) and 83 sorted S1 units (between 59–96) were recorded per session. For participant 2, an average of 80 sorted SMG units (between 69–92) and 81 sorted S1 units (between 61 – 101) were recorded per session. For participant 1, offline datasets were composed of 8 trials per word across ten sessions. Trials during which participant errors occurred were excluded. In total, between 156-159 trials per word were included, with a total of 1257 trials for offline analysis. Two experimental conditions were run, an Auditory cue condition and a Written cue condition. On four nonconsecutive session days, the Auditory cue task was run first, and on six nonconsecutive days, the Written cue task was run first. For online analysis, datasets were recorded on three different session days, for a total of 304 trials. Participant 2 offline dataset was composed of 16 trials per word using the written cue modality over nine sessions. Error trials were excluded. In total, between 142-144 trials per word were kept, with a total of 1145 trials for offline analysis. For online analysis, datasets were recorded on three session days, leading to a total of 448 online trials. |
| Sampling strategy | For offline and online experiments, sessions were repeated on different session days. On each session day, results were significantly above chance, demonstrating stable results. Chance was determined by randomizing trial labels 100 times, creating a null distribution. Number of sessions, session length and number of trials was maximized according to participant availability and stamina. |
| Data collection | Data were recorded with Blackrock Neuroport arrays. Blackrock NSP (Neural signal processor) system and headstages were used. Two researchers were present during all data collection, blindness was not applicable. |
| Timing | Data were collected between July 2021 and December 2022 for participant 1. For participant 2, data were recorded in January 2023. |
| Data exclusions | Data were not excluded from the analysis, except if participants were unable to continue experiments due to intense fatigue. |
| Non-participation | No participant dropped out. |
| Randomization | Participants were not allocated into experimental groups. In both participant we evaluated internal speech representation in the supramarginal gyrus. |

# Reporting for specific materials, systems and methods

We require information from authors about some types of materials, experimental systems and methods used in many studies. Here, indicate whether each material, system or method listed is relevant to your study. If you are not sure if a list item applies to your research, read the appropriate section before selecting a response.

## Materials & experimental systems

| n/a | Involved in the study |
|---|---|
| ☒ | ☐ Antibodies |
| ☒ | ☐ Eukaryotic cell lines |
| ☒ | ☐ Palaeontology and archaeology |
| ☒ | ☐ Animals and other organisms |
| ☐ | ☒ Clinical data |
| ☒ | ☐ Dual use research of concern |
| ☒ | ☐ Plants |

## Methods

| n/a | Involved in the study |
|---|---|
| ☒ | ☐ ChIP-seq |
| ☒ | ☐ Flow cytometry |
| ☒ | ☐ MRI-based neuroimaging |

## Clinical data

Policy information about clinical studies

All manuscripts should comply with the ICMJE guidelines for publication of clinical research and a completed CONSORT checklist must be included with all submissions.

| | |
|---|---|
| Clinical trial registration | NCT01964261 |
| Study protocol | The study protocol is not available to the public, but information is present at clinicaltrials.gov. |
| Data collection | The first participant was implanted in November 2016. Data were collected between July 2021 and December 2022. Participant 2 was implanted in October 2022. For participant 2, data were recorded in January 2023. Sessions were recorded at participants residence. |
| Outcomes | We hypothesized that neurons in the supramarginal gyrus modulate to internal speech processed. To evaluate our hypothesis, we recorded from groups of neurons (22–56 per session day for participant 1, 69–92 for participant 2), and evaluated tuning to internally spoken words using a linear regression analysis and a Kruskal Wallis test. Results were compared to a null distribution over 9 and 10 session days respectively. We further hypothesized that we could decode internal speech offline and in real time. Classification performances during internal and vocalized speech processes were evaluated by performing a LDA classifier and to comparing results to a null distribution that involved shuffling labels 100 (offline) or 1000 (online) times on each individual session day. |

