## [Peer Review File · Nature Human Behaviour]

Peer Review Information

Journal: Nature Human Behaviour

Manuscript Title: Representation of internal speech by single neurons in human supramarginal gyrus

Corresponding author name(s): Sarah K. Wandelt

Reviewer Comments & Decisions:

Decision Letter, initial version:

2nd July 2023

Dear Dr Wandelt,

Thank you once again for your manuscript, entitled "Representation of internal speech by single neurons in human supramarginal gyrus", and for your patience during the peer review process. I am communicating this decision to you instead of Dr Ariani, as he is currently out of the office.

Your Article has now been evaluated by 3 referees. You will see from their comments copied below that, although they find your work of considerable potential interest, they have raised quite substantial concerns. In light of these comments, we cannot accept the manuscript for publication, but would be interested in considering a revised version if you are willing and able to fully address reviewer and editorial concerns.

We hope you will find the referees' comments useful as you decide how to proceed. If you wish to submit a substantially revised manuscript, please bear in mind that we will be reluctant to approach the referees again in the absence of major revisions.

Specifically, you will see that Reviewer 2 raises substantial concerns regarding the validity of your work in light of the fact that high decoding accuracy is obtained based on brain activity in a very small part of SMG (without the involvement of other classical language areas) in only one of the two patients. These are fundamental concerns and for a successful revision, we would expect compelling additional evidence that demonstrates the validity of decoding from SMG. Additionally, both Reviewer 2 and Reviewer 3 ask you to present full - and mirroring - analyses and results for Participant 2. This is crucial for determining both validity as well as the reasons why decoding was not successful for this patient.

Finally, your revised manuscript must comply fully with our editorial policies and formatting requirements. Failure to do so will result in your manuscript being returned to you, which will delay its consideration. To assist you in this process, I have attached a checklist that lists all of our

requirements. If you have any questions about any of our policies or formatting, please don't hesitate to contact me.

If you wish to submit a suitably revised manuscript, we would hope to receive it within 4 months. I would be grateful if you could contact us as soon as possible if you foresee difficulties with meeting this target resubmission date.

- Include a "Response to the editors and reviewers" document detailing, point-by-point, how you addressed each editor and referee comment. If no action was taken to address a point, you must provide a compelling argument. When formatting this document, please respond to each reviewer comment individually, including the full text of the reviewer comment verbatim followed by your response to the individual point. This response will be used by the editors to evaluate your revision and sent back to the reviewers along with the revised manuscript.
- Highlight all changes made to your manuscript or provide us with a version that tracks changes.

[REDACTED]

Thank you for the opportunity to review your work. Please do not hesitate to contact me if you have any questions or would like to discuss the required revisions further.

Sincerely,

Stavroula Kousta

Stavroula Kousta, PhD
Chief Editor
Nature Human Behaviour

on behalf of

Giacomo Ariani, PhD
Associate Editor
Nature Human Behaviour

Reviewer expertise:

Reviewer #1: brain decoding/BMI of (inner) speech, intracranial recordings

Reviewer #2: brain decoding/BMI of (inner) speech, intracranial recordings

Reviewer #3: brain decoding/BMI of (inner) speech, intracranial recordings

REVIEWER COMMENTS:

Reviewer #1:

Remarks to the Author:

This manuscript presents convincing evidence for the encoding of covert and overt speech by the ensemble activity of single units in the left supramarginal gyrus (SMG), and further shows that inner speech activity can be decoded in real time in one participant to predict which word or pseudoword has been imagined among a set of 8 predefined items. Using limited data and a simple discrete LDA classifier, a rather high decoding accuracy can be achieved suggesting that SMG can be a relevant target region for the design of a speech BCI. Even if this result could not be replicated in the second participant, this study is important and original because it is to my knowledge the first achievement of a real-time speech BCI based on single unit recordings in the SMG. With the intention to further improve the manuscript, I would like to put forward the following comments:

- 1) Some key papers of the speech BCI literature are not cited, in particular the recent preprint by Willett and colleagues (<https://doi.org/10.1101/2023.01.21.524489>) and the seminal study by F. Guenther and colleagues reporting the first speech BCI demonstration in a paralyzed individual enable to speak (PLoS One 2009 4:e8218).
- 2) I am assuming that the cross-phase decoding presented in Figure 5A&B has been done for each type of cue separately (so that for example, the decoder built on the speech phases of the auditory cue trials is different from that built from the written cue trials). If this is indeed the case (please confirm in the text), then how do the different within-cue classifiers also generalize across cue types? In other words, how the classifiers trained on the different auditory cue trial phases perform on the different phases of the written cue trials, and vice versa?
- 3) For the online experiments, the "overt speech" phase has been replaced by a feedback phase displaying the predicted word. Analyzing offline these trials, how would a decoder trained on the cue phase (audio or written) perform on the "feedback" phase for either correct and incorrect trials?
- 4) As underlined by the authors, the SMG is known to be involved in phonological processing of speech. Were some neurons tuned to specific phonemes? For example, could the ensemble activity predict the time instants of phoneme /i/ present in several words? This could reinforce the first part of the paper exploring speech encoding by SMG.
- 5) The study shows that S1 neurons are modulated by vocalized speech but not inner speech. This result is used as a verification that the patient does not make any significant articulatory movements during the inner speech phases. As mentioned by the authors acoustic contamination can occur when using the Neuroport System. It would thus be important to verify that the S1 activity does not reflect such artifact. Indeed, this phenomenon typically occurs through badly plugged pins while everything is fine for the others. Thus, if present, contamination may occur only on some channels and not necessarily all. For example, playing the devil's advocate, if 2 patient cables were used for participant 1, one for the two S1 arrays and another for the SMG array, contamination could have occurred only

on S1 channels. It would thus be best to verify the level of contamination during the audio cues and the speech phases for the different sessions (connectors might indeed be well plugged one day and not another day).

6) Assuming there is no acoustic contamination, could the different words then be also decoded from the S1 population during the overt speech phase?

Minor points:

- 1) What are the genders of the participants and their respective language lateralization?
- 2) Where are the S1 arrays located with respect to the cortical anatomy?
- 3) It would be good to remind at the beginning of the results section that for the main experiment, the participants were instructed to internally say the cued word during the internal speech phase
- 4) Regression: please define the X variables
- 5) dPCA: please explain in more details what is exactly done
- 6) Methods dPCA, what does "As in the original manuscript" refer to?
- 7) Do panels B-E of Figure 1 correspond to the same or different units? The unit number could be provided for each panel to clarify.
- 8) The choice of the SMG compared to typical ventral motor/premotor areas for a speech BCI could be further discussed in the text.

Reviewer #2:

Remarks to the Author:

The main objective of this study is to investigate the decoding of internal (covert) speech using single-neuron recordings based on microelectrode arrays placed in the supramarginal gyrus (SMG). Two tetraplegic participants had microelectrode arrays implanted in the SMG and primary somatosensory cortex (S1) and performed internal and vocalized speech tasks involving eight words. The results of this study demonstrate that single-unit activity in SMG provided high accuracy for decoding internal speech in Participant 1, both in offline decoding (55%) and real-time online decoding (79%). However, the decoding accuracy of single-unit activity in SMG was not reliable for internal and vocalized speech in Participant 2.

Given the importance of accurately decoding internal speech in individuals who lack the ability to communicate with the outside world (such as patients with amyotrophic lateral sclerosis [ALS] and brain lesions), the findings of this study could have significant impacts on their quality of life. Additionally, this study contributes to the existing literature by presenting a successful case of high-accuracy internal speech decoding using single-unit activity in SMG.

One concern regarding this study is the lack of reporting on the low decoding accuracies for Participant 2 in both online and offline decoding. The absence of appropriate decoding accuracy in Participant 2 raises significant doubts about the validity of this study.

Another concern is that this study reports high decoding accuracy for both internal and vocalized speech production based on brain activity in a very small part of SMG (sampled by one iEEG electrode), without involving other classical language areas like Broca's area.

Please find my additional comments below.

1. Abstract:

a. Although the abstract mentions the inclusion of two participants, it only reports the high decoding accuracy of one participant. The authors should also provide the low decoding accuracy of Participant 2.

b. The authors should temper down the claim of novelty in this study, as previous research, such as Moses et al. (N Engl J Med, 2021), has already investigated internal speech and reported high accuracy. It is essential to acknowledge the existing literature and clarify the unique contributions of this study.

c. The authors should clarify that the electrode placement in the primary somatosensory cortex (S1) was in the arm and/or hand regions, not the lip/mouth region. This clarification is crucial because activation in the arm and/or hand regions of S1 is not typically associated with overt speech.

2. Method details:

a. Implants:

i. Figure S4B only shows the locations of the implanted electrodes in SMG. Please include the locations of the implanted electrodes in S1 for both participants.

ii. The reference (Armenta Salas et al., 2018) mentioned for pre-surgical planning, localization fMRI tasks, and surgical techniques only discusses the pre-surgical planning in one participant. It is unclear if the same plan was used for the second participant, and it does not address any pre-surgical mapping for localizing language cortex. Please clarify if the placement of electrodes in SMG was based on any language tasks and, if so, which language task(s) were used. This information is important because the electrodes placed in SMG for Participant 2 did not provide accurate decoding of speech.

iii. Since an electrode was placed in the premotor cortex (PMv), it is worth considering why the authors did not investigate the decoding capability of electrodes in this region for internal or vocalized speech. This area may have potential value in speech decoding.

b. Data collection:

i. Please add a section explicitly stating the number of recorded and used trials for each word in both offline and online decoding for both participants.

ii. The statement "For Participant 2, 72 sorted SMG units were recorded" prompts the question of whether you recorded and analyzed data in S1 for this participant.

iii. The sentence "One auditory cue task and one written cue task were recorded on 10 individual session days in Participant 1" needs clarification. Did you use all the data collected in 10 days for analysis, and how about Participant 2? Please provide this information.

c. Online task:

i. The submitted manuscript is missing Supplementary Video 1.

ii. The statement "The online task was run on three individual session days" requires clarification regarding the duration of each session.

3. Quantification and Statistical Analysis:

a. In the linear regression analysis, please remove the extra "+" sign from the equation.

b. Why did you only use the Kruskal-Wallis analysis in Participant 2? Please provide an explanation for this choice.

4. Discussion:

a. The statement "Furthermore, high decoding required only 24 seconds of training data per word" is mentioned but not supported by any results in the manuscript. Please provide relevant results to substantiate this claim.

b. The authors should cite previous literature to compare and contrast their findings regarding "strong evidence for internal speech processing at the single-neuron level in SMG."

c. It is intriguing that the authors observed face and lip activity in a predominantly S1 arm region during vocalized speech. While it remains unclear if any language task was administered by Armenta Salas et al. (2018), I wonder if their fMRI data detected any significant activity in the S1 arm region during overt speech tasks.

d. The statement "However, in certain contexts, we observed identical or better classification accuracy during internal speech than vocalized speech (Figure 4A, Online decoding)" is not supported by Figure 4A, as it does not show online decoding. Additionally, Figure 4 does not indicate better classification accuracy during internal speech than vocalized speech. Please revise this statement to align with the available data.

e. Similar to Participant 1, the authors should report the decoding accuracies for online and offline decoding in Participant 2 to provide a comprehensive analysis.

5. Acknowledgements: To comply with patient privacy, please refrain from stating the participants' initial names (FG and AN) if they are not pseudonyms or identifiers.

Reviewer #3:

Remarks to the Author:

The authors present single unit and multi-unit activity data recorded from multi-electrode arrays chronically implanted in the left supramarginal gyrus (SMG) of 2 volunteer tetraplegic participants and

additionally in the arm area of the left primary somatosensory cortex (S1) of 1 of the volunteer tetraplegic participants performing a single word language task. In the task, participants either hear a word in audio form (performed by 1 of the participants) or read a word in text form (performed by 1 of the participants), after which they are instructed to say the word internally first without vocalization in one time period, and the vocally say the word in a following time period after a brief delay preceding each period. The stimuli consist of 6 multi-syllable real English words and 2 multi-syllable pseudowords. The authors report that in the SMG of both patients single and multi-unit activity were activated for both internal and vocalized speech. In the patient who performed both the audio and written versions of the task, the responses in the word presentation time period differentiated between whether the word was presented in audio or text form, but the responses in both the internal and vocalized speech versions of the task did not differentiate between the modality of the word cue. In that patient, the SMG unit responses to different words were fed to both offline and online classification algorithms that were able to decode which of the 8 words were presented and repeated by the participant on a given trials with good success. This included successful decoding during the internal speech period, providing an important proof-of-principle for potential future language brain-machine interface (BMI) applications in the SMG. Follow-up analyses suggested largely similar codes in terms of which units prefer which words between the internal and vocalized speech periods. In contrast, activity in the S1 of the same patient was only active during the speech vocalization time period, which the authors attribute to the existence of face and lip sensory activity that there is previous evidence of existing within the arm area of S1. An additional control experiment in one patient with a subset of words suggested that both a sound-based internal speech strategy and internalized visualization of the text representation of a word yielded similar success rates in decoding in the SMG.

This manuscript presents unique and important data for both the BMI and neuroscience of language fields. The inclusion of pseudo-words in the experimental stimuli provides some important insight suggesting that the language signals in the SMG during this task may likely be phonetic in nature. The cross-phase validation analyses that the authors employ show that the neural code in this area is largely preserved through-out word perception responses, internal speech and vocalized speech.

With that said, there are a number of issues to point out in the present manuscript, all of which are addressable.

The authors should attach significance testing to their Figure 3 Demixed PCA analyses or otherwise show the temporal trajectory of significant word decoding identity in their data. For publication in *Nature Human Behavior*, that is not an unreasonable request, as similar articles in this space, including from this group (e.g. Aflalo et al. 2020) have done this. There are also issues with their execution and description of the Demixed PCA analysis, which I describe later on in these comments.

The authors mention that due to software constraints, the auditory cue appeared on average 250 ms later than the written cue, although the 2 single trial examples in Figure 1 suggest the difference is closer to ~750 ms. These things happen in the course of conducting experiments, but unfortunately it does complicate the inferences that can be gleaned from the cue period, which the present manuscript has not adequately addressed. The authors should account for this onset differences in their analyses and in the figures showing the time course in the cue period (Figure 2A, Figure 3, Figure S2A). Ideally this should be done on a word-per-word basis, as altogether, this comes across as though there was some unintended variability in the onset time of the audio signal. With regards to this issue and Figure

2A, right now the most striking difference between the auditory and written cues in Figure 2A that a reader would notice is the latency difference between the two, which the authors don't discuss. So the issue needs to be addressed. The neural difference between the auditory and written cue responses show a latency difference that seems closer to the ~750 ms number from the examples in Figure 1.

There is some sloppiness to the writing. For example at the bottom of page 21, whatever study that sentence was supposed to reference is not there. The writing comes across as being written by a non-native English speaker. There are a number of examples of the wrong preposition being used, which is admittedly not easy to master in English. As one example of unclarity in the writing, in the last paragraph of page 10, for the sentence: "These results were significantly higher for internal speech and vocalized speech" It is not possible for a reader to determine what this refers to. Probably Cue, Internal, and Speech periods were compared to the ITI period, although in the context of this paragraph the text itself suggests Cue, Internal, and Speech conditions were compared to each other. The next version of the manuscript should be edited in that regard by a native English speaker before it's submitted.

It would be considerably helpful if the authors could include single neuron examples of neuronal activity traces that discriminate between words, to give the reader a better sense of what is driving the word decoding results.

I suspect the following might not have been done due to sensitivity concerns, but it bears mentioning here that including the location of the implants in Figure 1 rather than Figure S4B would be helpful scientifically if the authors are able to do it. The precise location of these implants are of course of great interest to the scientists interacting with this work.

I was amazed that the text in the article does not mention anywhere that the array is in the left hemisphere. This detail is important for all of the implications of this study. As it stands, one can only gather that this from the orientation of the brain in Figure S4B.

The authors' use of the word "tuning" in this manuscript does not match the analyses that they conducted. For example, in the visual system, to say that a neuron is "tuned" to orientation means that it responds preferentially to bars of light of certain orientations but not of other orientations. In this manuscript, tuning for words would reflect significant preference for one word over another word. However in the analysis the authors conducted, a unit was considered tuned if its response to at least one word was significantly larger than its ITI response. Thus a neurons that responded to every word equally would be considered tuned. The authors should change either the label they use or their analyses to assess tuning. For example a significant ANOVA or Kruskal-Wallis test across words would show significant word-tuning as the word is typically used.

In the authors' Demixed PCA analysis, they first state that they followed the method described in Kobak et al. 2016, but the details that they describe after that differs from what Kobak et al. 2016 describe, in a manner such that the way that the what the present authors did differs from what Kobak et al. 2016 did does not appear to be sensible. Briefly, the 8 terms of the ANOVA-like decomposition in Kobak et al. 2016 are grouped into 5 terms by grouping e.g. the stimulus term with the stimulus-time interaction term, with the time term appearing alone. Here the authors apparently combined the time term with cue modality-word-time interaction to get their ultimate "Timing" factor, which makes no sense. If the authors follow Kobak et al. 2016 they would have a fourth type of component for cue modality-word interactions. Following from the confusion with the analysis, the

authors' description of this analysis in the methods is not self-consistent, so care is needed both for this analysis itself and the descriptions of this analysis in the next version of the manuscript. Some other details to point out, the authors state that individual components are synonymous with marginalizations, which is not correct. They state that ANOVA involves covariance decomposition when it involves variance decomposition. Finally, it would be helpful for the reader unfamiliar with the technique to specify in the results text and caption that Figure 3 shows the projection of the data onto the respective dPCA decoder axis.

It is unnecessarily difficult for the reader to discern which results are from which participant. The authors should either add a blanket statement in the beginning of the methods that "unless indicated otherwise, results are for patient 1" or say individually for each figure e.g. "this shows data from participant 1". It is also strange that the Kruskal-Wallis analysis was done on participant 2's data, but not participant 1's data. Ideally a Kruskal-Wallis analysis of participant 1's data would be a better estimate of tuning in a neuron rather than the t-test that the authors employed. Additionally, the inclusion of both participant 2's data and the S1 data both strengthen this manuscript overall, however the reader is left to wonder why the S1 data from Patient 2 was not analyzed. The authors should either include analyses of these data, or explain why they are not doing that.

The introduction would benefit from a paragraph motivating the author's decision to include pseudowords in their stimuli. Similarly they should in the introduction briefly motivate the reason to include S1. The authors mention that they hypothesize that S1 would modulate in the speech period only, but there is no explanation at all of what that hypothesis is based on in the introduction, although it does come in the discussion section later.

In the linear regression analysis section of the methods, there is a "++" typo in the equation, and I would recommend subscripts for the numbers rather than "B1X1" if that equation is ultimately included. However, I don't think the description of this as a linear regression is appropriate- as the authors' text describes, what they performed are 8 t-tests comparing the activity to each ITI period. This would be more clearly described as 8 t-tests with a pooled ITI baseline term. Moreover the t-tests used are describe as 'student's t-tests' which generally refer to unpaired, 2 sample t-tests. Paired t-tests are the correct tests to use in this instance.

The alpha level for the Kruskal-Wallis test applied to participant 2's data is not mentioned, which becomes particularly important detail as the authors find proportions near %5 with significant results, which they state reflects a significant proportion of their recorded population. They authors need to state the alpha they used for this test, and follow it up with a binomial test in order to conclude that there are a significant proportion of tuned neurons in their population.

The details of the authors' classification analyses are under-specified. They describe using a training and testing set, but don't mention how they partitioned the data into these 2 sets, aside from saying that they performed leave one out validation. This language is unnecessarily unclear- they should just say that they used a leave one out method. They should describe what the one item left out was - one neurons vs. one trial across neurons vs. one trial for one neuron. For the cross-phase classification, the authors' say they performed that following the details of the intra-phase classification, however how that would occur is not at all clear- they need to specify what was left out in the cross-phase classification and tested across iterations. The alpha level of 0.001 for the test is unusually low, which if used requires explanation.

There are some unspecified details that should be added to the methods section. The authors should say in the methods how many trials were in the session, and the randomization and balancing procedure used, and the order in which the tasks were performed (auditory cue vs. written cue) in participant 1 in a given day.

For Figure 5, for testing the proportions of cells with a binary feature like tuned vs. not-tuned, Fisher's exact test would be the appropriate test. In the last sentence of page 17, the authors seem to be describing a comparison between the second-most right bars in the plots of 5B versus the same in 5A. However that comparison does not seem like it would be significant. Please clarify this.

Author Rebuttal to Initial comments

Reviewer guide:

Responses to reviewers: **bold**

Manuscript text: **blue**

New edits to manuscript text: **bold blue**

REVIEWER COMMENTS:

Reviewer #1:

Remarks to the Author:

This manuscript presents convincing evidence for the encoding of covert and overt speech by the ensemble activity of single units in the left supramarginal gyrus (SMG), and further shows that inner speech activity can be decoded in real time in one participant to predict which word or pseudoword has been imagined among a set of 8 predefined items. Using limited data and a simple discrete LDA classifier, a rather high decoding accuracy can be achieved suggesting that SMG can be a relevant target region for the design of a speech BCI. Even if this result could not be replicated in the second participant, this study is important and original because it is to my knowledge the first achievement of a real-time speech BCI based on single unit recordings in the SMG. With the intention to further improve the manuscript, I would like to put forward the following comments:

We thank the reviewer for the thorough assessment of our manuscript, and are pleased that our main findings came across clearly. We have added the additional analysis in the figures as recommended and written text to accompany each. Below, we have also addressed each comment point by point.

1) Some key papers of the speech BCI literature are not cited, in particular the recent preprint by Willett and colleagues (<https://doi.org/10.1101/2023.01.21.524489>) and the seminal study by F. Guenther and colleagues reporting the first speech BCI demonstration in a paralyzed individual enable to speak (PLoS One 2009 4:e8218).

We thank the reviewer for suggesting the addition of the following papers. The papers were included in the manuscript.

However, vocalized speech has also been decoded from localized regions of the cortex. Using a neurotrophic electrode, (Guenther et al. 2009), demonstrated in 2009 real-time speech synthesis from the motor cortex. More recently, speech neuroprosthetics were built from small-scale microelectrode arrays located in the motor cortex (Stavisky et al. 2019; Wilson et al. 2020), premotor cortex (Willett et al. 2023) and the supramarginal gyrus (SMG) (Wandelt et al. 2022), demonstrating vocalized speech BMIs can be built using neural signals from localized regions of cortex.

2) I am assuming that the cross-phase decoding presented in Figure 6A&B has been done for each type of cue separately (so that for example, the decoder built on the speech phases of the auditory cue trials is different from that built from the written cue trials). If this is indeed the case (please confirm in the text), then how do the different within-cue classifiers also generalize across cue types? In other words, how the classifiers trained on the different auditory cue trial phases perform on the different phases of the written cue trials, and vice versa?

We thank the reviewers for bringing up this valuable additional analysis. In Figure 6, models were built separately for written and auditory cue trials. A clarification has been added to the text.

To explore the relationships between each of these processes, we used cross-phase classification to identify the distinct and common neural codes **separately in the auditory and written cue datasets.**

Additionally, we have computed a cross-decoding analysis, utilizing auditory trial data to decode written cue trial data and vice-versa. We found significant cross-decoding outcomes spanning across the cue, internal, and vocalized speech phases. These results underscore the robustness and consistency of cue independent internal and vocalized speech representation in SMG, confirming findings of the dPCA analysis (Figure 3 – Cue modality).

3) For the online experiments, the “overt speech” phase has been replaced by a feedback phase displaying the predicted word. Analyzing offline these trials, how would a decoder trained on the cue phase (audio or written) perform on the “feedback” phase for either correct and incorrect trials?

We thank the reviewers for suggesting this interesting additional analysis. For online trials, words were cued exclusively with written cues. To perform this analysis, we aggregated trials from all online sessions and trained a decoder during the cue phase to predict the feedback phase. Subsequently, we computed the percentage of accurately classified trials, differentiating between those that were correct and those that were incorrect. As anticipated, our findings demonstrated that a decoder

trained on the cue phase exhibited greater accuracy in correct trials (during which the word on the screen was identical to the one during the cue phase), than during misclassified trials (during which the word on the screen was not identical to the one during the cue phase).

4) As underlined by the authors, the SMG is known to be involved in phonological processing of speech. Were some neurons tuned to specific phonemes? For example, could the ensemble activity predict the time instants of phoneme /i/ present in several words? This could reinforce the first part of the paper exploring speech encoding by SMG.

We appreciate the reviewer's insightful suggestion regarding the involvement of phonemes in SMG. Accurate identification of phonemes that occur in different words would require aligning the neural data to word onset. Our current study primarily focuses on internal speech, and this additional analysis would require a substantially different task (or possibility several) to allow for such

alignment. We intend to conduct controlled experiments in the future that will allow us to study this question more comprehensively for a subsequent manuscript.

5) The study shows that S1 neurons are modulated by vocalized speech but not inner speech. This result is used as a verification that the patient does not make any significant articulatory movements during the inner speech phases. As mentioned by the authors acoustic contamination can occur when using the Neuroport System. It would thus be important to verify that the S1 activity does not reflect such artifact. Indeed, this phenomenon typically occurs through badly plugged pins while everything is fine for the others. Thus, if present, contamination may occur only on some channels and not necessarily all. For example, playing the devil's advocate, if 2 patient cables were used for participant 1, one for the two S1 arrays and another for the SMG array, contamination could have occurred only on S1 channels. It would thus be best to verify the level of contamination during the audio cues and the speech phases for the different sessions (connectors might indeed be well plugged one day and not another day).

We appreciate the reviewers' observation regarding the possibility of contamination occurring solely on S1 channels due to the use of different patient cables. While we acknowledge the potential for auditory contamination, we maintain that its presence would manifest not only during the vocalized speech phase but also during the auditory cue phase.

Figure S2, Panel A (first submission manuscript, reviewed manuscript Figure S3, Panel A), in our manuscript clearly illustrates the absence of significant word tuning in S1 during the cue phase, regardless of whether the cue was auditory or written. Moreover, as depicted in Figure S2, Panel B, no noteworthy word decoding is observed in S1 during the cue phase. These findings provide compelling evidence that contamination, if it were to occur, should be evident during the auditory cue phase as well. Therefore, we are confident no contamination artificially inflated decoding accuracies in our neural data.

To address the reviewer's concern more comprehensively, we have computed the figures below in the response to the reviewers. The first figure shows the percentage of tuned neurons in S1 for each session day individually, with an example audio trial in participant 1. As can be seen, little to no word tuning is observed during the cue phase.

S1 tuning for individual session days - Audio cue trials

The second figure shows S1 offline classification accuracies, only including auditory cue trials (in contrast to the figure in the manuscript that includes both auditory and written cue trials). Here again, we do not observe significant decoding during the cue phase.

Together, these figures suggest no audio contamination was observed in S1.

These changes are reflected in the text:

“However, no significant classification was detected in S1 channels during the auditory cue phase nor the vocalized speech phase (Figure S2,B).”

6) Assuming there is no acoustic contamination, could the different words then be also decoded from the S1 population during the overt speech phase?

As can be seen in Figure S2, Panel B (Figure S4, panel A,C), while average decoding is above chance level during vocalized speech in S1, decoding did not reach significance. These findings are in concordance with a recent imaging study performed by (Muret et al. 2022). In this fMRI study, lip and face activity could be detected in S1 leg and hand region of interests (ROIs). However, differentiation (decoding) of two lip actions was only found in S1 face ROI.

Minor points:

1) What are the genders of the participants and their respective language lateralization?

We thank the reviewer for bringing this to our attention. Both participants in our study were male, and the implants were situated on the left hemisphere. The manuscript text was adapted accordingly.

Two **male** tetraplegic participants were recruited for an IRB- and FDA-approved clinical trial of a brain-machine interface and gave informed consent to participate.

2) Where are the S1 arrays located with respect to the cortical anatomy?

We thank the reviewer for bringing this to our attention. All implants were located on the left hemisphere. The manuscript text was adapted accordingly. Additionally, Figure 2 was added to the manuscript depicting implant locations.

Data were collected from implants located in the **left** supramarginal gyrus (SMG) and the **left** primary somatosensory cortex (S1) (for anatomical locations see **Figure 2**).

3) It would be good to remind at the beginning of the results section that for the main experiment, the participants were instructed to internally say the cued word during the internal speech phase

We thank the reviewer for suggesting this clarification. The manuscript was adapted accordingly.

“The participant was instructed to internally say the cued word during the internal speech phase, and to vocalize the same word during the speech phase.”

4) Regression: please define the X variables

The methods were adapted in the following way:

$$FR = \sum_{w=1}^W \beta_w X_w + \beta_0$$

where FR corresponds to the firing rate of the unit, β_0 to the offset term equal to the average ITI data of the unit, **X is the vector indicator variable for each word w** , and β_w corresponds to the estimated regression coefficient for word w . W was equal to 8 (Battlefield, Cowboy, Python, Spoon, Swimming, Telephone, Bindip, Nifzig). (see Wandelt et al. 2022).

5) dPCA: please explain in more details what is exactly done

6) Methods dPCA, what does “As in the original manuscript” refer to?

For question 5) and 6). The dPCA method section was updated. See text below.

dPCA was performed on the session data to study the activity of the neuronal population in relation to the external task parameters: cue modality and word. Kobak et al (2016) introduced dPCA as a refinement of their earlier dimensionality reduction technique (of the same name) which attempted to combine the explanatory strengths of linear discriminant analysis (LDA) and principal component analysis (PCA). By deconstructing neuronal population activity into individual components, each component related to a single task parameter (code: github.com/machenslab/dPCA).

This text follows the methodology outlined by Kobak et al. (2016). Briefly, this involved the following steps for N neurons:

First, unlike in PCA, we focused not on the matrix, X , of the original data, but on the matrices of marginalizations, X_ϕ . The marginalizations were computed as neural activity averaged over trials, k , and some task parameters in analogy to the covariance decomposition done in MANOVA. Since our dataset has three parameters: timing, t , cue modality, c (eg. auditory or visual) and word, w (8 different words), we obtained the total activity as the sum of the average activity with the marginalizations and a final noise term

$$X_{tcwk} = \underline{X} + \underline{X}_t + \underline{X}_c + \underline{X}_{tc} + \underline{X}_w + \underline{X}_{tw} + \underline{X}_{cw} + \underline{X}_{tcw} + \epsilon_{tcwk}$$

The above notation of Kobak et al is the same as used in factorial ANOVA, i.e. X_{tcwk} = matrix of firing rates for all neurons, $\langle \cdot \rangle_{ab}$ = average over a set of parameters a, b, \dots , $\underline{X} = \langle X_{tcwk} \rangle_{tcwk}$, $\underline{X}_t = \langle X_{tcwk} - \underline{X} \rangle_{cw}$, $\underline{X}_{tc} = \langle X_{tcwk} - \underline{X} - \underline{X}_t - \underline{X}_c - \underline{X}_w \rangle_{wk}$ and so on. Finally, $\epsilon_{tcwk} = X_{tcwk} - \langle X_{tcwk} \rangle_k$.

Participant 1 datasets were composed of $N = 333$ (SMG), $N = 828$ (S1) and $k = 8$. Participant 2 datasets were composed of $N = 547$ (SMG), $N = 522$ (S1) and $k = 16$. To create balanced datasets, error trials were replaced by the average firing rate of $k-1$ trials.

Our second step reduced the number of terms by grouping them as seen by the braces in the equation above, since there is no benefit in demixing a time-independent pure task, a , term \underline{X}_a from the time-task interaction terms \underline{X}_{ta} since all components are expected to change with time. The above grouping reduced the parametrization down to just five marginalization terms and the noise term (reading in order): the mean firing rate, the task-independent term, the cue modality term, the word term, the cue modality-word interaction term and the trial-to-trial noise.

Finally, we gained extra flexibility by having two separate linear mappings F_ϕ for encoding and D_ϕ for decoding (unlike in PCA, they are not assumed to be transposes of each other). These matrices were chosen in order to minimize the loss function (with a quadratic penalty added to avoid overfitting):

$$L_\phi = \|X_\phi - F_\phi D_\phi X\|^2 + \mu \|F_\phi D_\phi\|^2$$

Here, $\mu = (\lambda \|X\|)^2$ where λ was optimally selected through 10-fold cross-validation in each dataset.

We refer the reader to Kobak et al. for a description of the full analytic solution.

7) Do panels B-E of Figure 2 correspond to the same or different units? The unit number could be provided for each panel to clarify.

Separate neurons were used for each example. We added supplementary figure S2 in the manuscript, showing neuronal responses to all words for participant 1 and participant 2, with the corresponding unit number. Specifically, panel A and B show the response of the same unit to Auditory cue trials (left) and Written cue trials (right) on the same session day.

Figure S2 | SMG firing rates for eight words over trial duration. Example smoothed firing rates of neurons tuned to eight words in SMG for participant 1 (A-C) and participant 2 (D). Figures shows the average firing rate over eight trials (solid line: mean, shaded area: 95% bootstrapped confidence interval) starting 1 second before cue presentation. A) and B) show tuning of the same neuron on the same day in the “Auditory cue” and the “Written cue” task, demonstrating stable word representation in different task conditions.

8) The choice of the SMG compared to typical ventral motor/premotor areas for a speech BCI could be further discussed in the text.

The rationale behind employing the SMG for investigating internal speech arises from our previous study (Wandelt et al. 2022), where we identified vocalized speech representation in this region. We added a sentence to emphasize that information in the introduction:

In this work, we build on our prior research which identified vocalized speech representation within SMG (Wandelt et al. 2022), to discern whether it also encodes internal speech.

It's important to note that the localization was not specifically conducted for speech-related purposes. We mention that aspect in the methods:

Placement of electrodes was based on fMRI tasks involving grasp and dexterous hand movements.

And in the discussion:

Implant locations for both participants were informed from pre-surgical anatomical / vasculature scans and fMRI tasks designed to evoke activity related to grasp and dexterous hand movements (see Armenta Salas et al. 2018). [...] A pre-surgical assessment of functional activity related to language and speech may be required to determine the best candidate implant locations within SMG for online speech decoding applications.

Reviewer #2:

Remarks to the Author:

The main objective of this study is to investigate the decoding of internal (covert) speech using single-neuron recordings based on microelectrode arrays placed in the supramarginal gyrus (SMG). Two tetraplegic participants had microelectrode arrays implanted in the SMG and primary somatosensory cortex (S1) and performed internal and vocalized speech tasks involving eight words. The results of this study demonstrate that single-unit activity in SMG provided high accuracy for decoding internal speech in Participant 1, both in offline decoding (55%) and real-time online decoding (79%). However, the decoding accuracy of single-unit activity in SMG was not reliable for internal and vocalized speech in Participant 2.

Given the importance of accurately decoding internal speech in individuals who lack the ability to communicate with the outside world (such as patients with amyotrophic lateral sclerosis [ALS] and brain

lesions), the findings of this study could have significant impacts on their quality of life. Additionally, this study contributes to the existing literature by presenting a successful case of high-accuracy internal speech decoding using single-unit activity in SMG.

One concern regarding this study is the lack of reporting on the low decoding accuracies for Participant 2 in both online and offline decoding. The absence of appropriate decoding accuracy in Participant 2 raises significant doubts about the validity of this study.

Another concern is that this study reports high decoding accuracy for both internal and vocalized speech production based on brain activity in a very small part of SMG (sampled by one iEEG electrode), without involving other classical language areas like Broca's area.

We acknowledge the reviewer's comments and would like to express our gratitude for their valuable input. In response, we have increased the number of sessions for participant 2, mirroring the analysis in participant 1. A recent study from Willett et al. (Willett et al. 2023) suggests that Broca's area may not be an ideal target region for speech decoding applications. In their study, four microelectrode arrays were implanted in the ventral premotor cortex and two in area 44 (part of Broca's area). However, their implant in Broca area weakly encoded words and phonemes and was subsequently not included as input for online speech decoding. These results are consistent with recent findings that damage in left Broca's area may not contribute to long-term speech production outcome after stroke (Gajardo-Vidal et al. 2021). Below, we have also addressed each comment point by point.

Please find my additional comments below.

1. Abstract:

a. Although the abstract mentions the inclusion of two participants, it only reports the high decoding accuracy of one participant. The authors should also provide the low decoding accuracy of Participant 2.

The abstract was edited to include the results of participant 2.

In this work, two tetraplegic participants with implanted microelectrode arrays located in the supramarginal gyrus (SMG) and primary somatosensory cortex (S1) performed internal and vocalized

speech of six words and two pseudowords. In both participants, we found significant neural representation of internal and vocalized speech, at the single neuron and population level in SMG. From recorded population activity in SMG, the internally spoken and vocalized words were significantly decodable. In an offline analysis, we achieved average decoding accuracies of 55% and 24% for each participant, respectively (chance level 12.5%) and during an online internal speech BMI task, we averaged 79% and 23% accuracy, respectively.

b. The authors should temper down the claim of novelty in this study, as previous research, such as Moses et al. (N Engl J Med, 2021), has already investigated internal speech and reported high accuracy. It is essential to acknowledge the existing literature and clarify the unique contributions of this study.

We agree that it is important to acknowledge existing literature. However, we wish to address potential confusion regarding the focus of Moses et al.'s investigation and stand by the claim of novelty in our study. Moses et al. (2021) decoded attempted speech, which encompasses both auditory and movement components. In a more recent study from the same lab, Metzger et al. (2022) investigated silent speech (the mouthing of words), which requires articulating the word. In our study, the patient engaged in a fully internalized experiment, devoid of any motor involvement. This unique approach eliminates the necessity for motor components and would be promising for individuals who may be entirely locked in. These distinctions are covered in the introduction of the manuscript.

While important advances in vocalized speech (Makin, Moses, and Chang 2020), attempted speech (Moses et al. 2021), and mimed speech (Bocquelet et al. 2016; Anumanchipalli, Chartier, and Chang 2019; Metzger et al. 2022; Willett et al. 2023; Metzger et al. 2023) decoding have been made, highly accurate internal speech decoding has not been achieved. Lack of behavioral output, lower signal-to-noise ratio, and differences in cortical activations compared to vocalized speech are speculated to contribute to lower classification accuracies of internal speech (Angrick et al. 2018; Martin et al. 2018; Luo, Rabbani, and Crone 2022; Proix et al. 2022; Meng et al. 2023).

c. The authors should clarify that the electrode placement in the primary somatosensory cortex (S1) was in the arm and/or hand regions, not the lip/mouth region. This clarification is crucial because activation in the arm and/or hand regions of S1 is not typically associated with overt speech.

Thank you for pointing out the reference to the electrode placement in our manuscript. This detail has been appropriately addressed in the text but we acknowledge that it should be clarified in the method section as well.

Correspondingly, we observed generalized face and lip activity in a predominantly S1 arm region (as detailed in Armenta Salas et al., 2018 for implant location) during vocalized speech.

We added the following edits (in bold) in the discussion.

The first participant underwent surgery in November 2016 to implant two 96-channel multi-electrode arrays (Neuroport Array, Blackrock Microsystems, Salt Lake City, UT) in SMG and in the ventral premotor cortex (PMv) and two 7 x 7 sputtered iridium oxide film (SIROF) - tipped microelectrode arrays with 48 channels each in **the hand and arm area of S1**.

2. Method details:

a. Implants:

i. Figure S4B only shows the locations of the implanted electrodes in SMG. Please include the locations of the implanted electrodes in S1 for both participants.

We added participant 1 and participant 2 SMG and S1 implant locations as Figure 1 in the manuscript.

Figure 1 | Multielectrode implant locations. **A)** SMG implant locations in participant 1 (1 x 96 multielectrode array) **B)** and participant 2 (1 x 64 multielectrode array). **C)** S1 implant locations in participant 1 (2 x 96 multielectrode array) and **D)** participant 2 (2 x 64 multielectrode array).

ii. The reference (Armenta Salas et al., 2018) mentioned for pre-surgical planning, localization fMRI tasks, and surgical techniques only discusses the pre-surgical planning in one participant. It is unclear if the same plan was used for the second participant, and it does not address any pre-surgical mapping for localizing language cortex. Please clarify if the placement of electrodes in SMG was based on any language tasks and, if so, which language task(s) were used. This information is important because the electrodes placed in SMG for Participant 2 did not provide accurate decoding of speech.

Thank you for pointing out the reference to the pre-surgical planning. This detail has been addressed in the discussion, but should be clarified in the method section as well.

Discussion: Implant locations for both participants were informed from pre-surgical anatomical / vasculature scans and fMRI tasks designed to evoke activity related to grasp and dexterous hand movements (see Armenta Salas et al. 2018). Furthermore, the number of electrodes of the implanted array was higher in the first participant (96) than in the second participant (64) (Figure S4B). A pre-surgical assessment of functional activity related to language and speech may be required to determine the best candidate implant locations within SMG for online speech decoding applications.

We added the following text in the methods:

Placement of electrodes was based on fMRI tasks involving grasp and dexterous hand movements.

iii. Since an electrode was placed in the premotor cortex (PMv), it is worth considering why the authors did not investigate the decoding capability of electrodes in this region for internal or vocalized speech. This area may have potential value in speech decoding.

We thank the reviewer for their suggestion to investigate decoding capabilities in the premotor cortex (PMv). In our study, we encountered limitations in obtaining a substantial number of neurons from the PMv region, which subsequently impacted the quality of our results. This issue aligns with findings presented in Wandelt et al. (2022), where similar challenges were highlighted in obtaining robust PMv activity. Given the lack of discernable neurons, we have refrained from pursuing this avenue of investigation for internal or vocalized speech decoding.

b. Data collection:

i. Please add a section explicitly stating the number of recorded and used trials for each word in both offline and online decoding for both participants.

We added a section in the methods explaining the number of recorded and used trials for each word in both offline and online decoding.

Total number of recording trials

For participant 1, we collected offline datasets composed of 8 trials per word for each cue across ten sessions. Trials during which participant errors occurred were excluded. In total, between 156-159 trials per word were included, leading to a total of 1257 trials for offline analysis. On four nonconsecutive session days, the Auditory cue task was run first, and on six nonconsecutive days, the Written cue task was run first. For online analysis, datasets were recorded on three different session days, for a total of 304 trials. Participant 2 underwent a similar data collection process, with offline datasets comprising 16 trials per word using the written cue modality over nine sessions. Error trials were excluded. In total, between 142-144 trials per word were kept, leading to a total of 1145 trials for offline analysis. For online analysis, datasets were recorded on three session days, leading to a total of 448 online trials.

ii. The statement "For Participant 2, 72 sorted SMG units were recorded" prompts the question of whether you recorded and analyzed data in S1 for this participant.

S1 analysis data for participant 2 was added in the manuscript. Specifically, S1 dPCA analysis was performed (Figure 3, panel D,E, right), tuning in 50ms bins (Figure S3, panel B), and decoding analysis (Figure S4, panel C).

Figure 5 | Demixed principal component analysis (dPCA) highlights SMG’s involvement in language processing. dPCA was performed to investigate variance within three marginalizations: “Timing”, “Cue Modality”, and “Word” for participant 1 (A-C) and “Timing” and “Word” for participant 2 (D-E). Demixed principal components explaining the highest variance within each marginalization were plotted over time, by projecting the data onto their respective dPCA decoder axis. **A)** The “Timing” marginalization demonstrates SMG modulation during cue, internal speech and vocalized speech, while S1 only represents vocalized speech. **B)** The “Cue Modality” marginalization suggests internal and vocalized speech representation in SMG are not affected by the cue modality. **C)** The “Word” marginalization shows high variability for different words in SMG, but near zero for S1. **D)** Same as A) for participant 2. **E)** Same as C) for participant 2. Variance for different words in SMG (left) was higher than in S1 (right), but lower in comparison to SMG in participant 1 (C).

Figure S3 | Tuning analysis demonstrate selective word representation in SMG but no internal word activation in S1. **A)** Participant 1 S1 tuning analysis through linear regression. Average percentage of tuned neurons to words in 50ms time bins in S1 over the trial duration for “Auditory cue” (blue) and “Written cue” (green) tasks (solid line: mean over 10 sessions, shaded area: 95% confidence interval). **B)** Same as A) for participant 2, with 9 sessions. Tuning during internal speech was significantly higher than

during ITI (paired t-test, $p = 0.016$). These results show while lip and face activity are represented in the putative arm area in S1, no activity is elicited during internal speech.

Figure S4 | S1 and SMG offline decoding accuracies. A) Participant 1 S1 decoding analysis. “Auditory cue” and “Written cue” tasks data were combined for each individual session day (~16 trials per word) and leave one out cross-validation was performed (black dots). PCA was performed on the training data, a LDA model was constructed, and results were plotted with 95% c.i. of the session means. Significance of classification accuracies was evaluated by comparing results to a shuffled distribution (averaged shuffle results = red dots). No classification accuracy was significant. However, classification accuracy during vocalized speech was significantly higher than during the previous delay period (paired t-test: * $p < 0.05$). Lack of decoding during the cue phase suggest no auditory contamination occurred in S1 channels. **B)** For participant 2, data of 16 trials per word during the “Written cue” experiment were combined. In SMG, significant word decoding was observed during the cue, internal and vocalized speech phases (null distribution, turquoise - mean decoding value above 97.5 / 99.5 percentile of shuffle distribution = */**). Decoding accuracies were significantly higher in the cue and internal speech condition, compared to rest phases ITI and D1 (paired t-test, black - $p < 0.05$ / $0.01 = */ **$). **C)** S1 decoding mirrored results in participant 1, suggesting no synchronized face movements occurred during the cue phase or internal speech phase.

The manuscript was adapted accordingly.

For participant 1, an average of 33 sorted SMG units (between 22–56) and 83 sorted S1 units (between 59–96) were recorded per session. **For participant 2, an average of 80 sorted SMG units (between 69–92) and 81 sorted S1 units (between 61 – 101) were recorded per session.**

iii. The sentence "One auditory cue task and one written cue task were recorded on 10 individual session days in Participant 1" needs clarification. Did you use all the data collected in 10 days for analysis, and how about Participant 2? Please provide this information.

All the data recorded on the 10 individual session days were used for analysis. For participant 2, additional recordings were performed. In total, 9 offline session blocks and 3 online session blocks were recorded for participant 2. For more information about number of trials included in the analysis, see our response to comment b. Data collection: i. above.

c. Online task:

i. The submitted manuscript is missing Supplementary Video 1.

Add supplementary video 1 to submission.

ii. The statement "The online task was run on three individual session days" requires clarification regarding the duration of each session.

The online sessions were conducted across three distinct session days, each encompassing several datablocks with varying trial counts. The total aggregate number of trials has been provided for reference above in response of comment b.i. Our approach involved retraining the decoder after each run, leading to distinct model iterations for different online trials. As a consequence of this iterative process, we have represented the results individually. We clarified the total number of online trials in the method (see the response to comment b. Data collection: i. above).

3. Quantification and Statistical Analysis:

a. In the linear regression analysis, please remove the extra "+" sign from the equation.

We thank the reviewer for noticing the extra + sign in the equation. The equation has been replaced by the following formula in the manuscript.

$$FR = \sum_{w=1}^W \beta_w X_w + \beta_0$$

b. Why did you only use the Kruskal-Wallis analysis in Participant 2? Please provide an explanation for this choice.

We utilized the Kruskal-Wallis analysis for Participant 2 due to its omnibus nature, emphasizing neurons that have the power to discriminate between different class conditions. In the new version of the manuscript, we have incorporated both methods for both participants in supplementary figure S3. This approach provides a comprehensive view of our findings. Please see figures as response to reviewer 3, comment 7 below.

4. Discussion:

a. The statement "Furthermore, high decoding required only 24 seconds of training data per word" is mentioned but not supported by any results in the manuscript. Please provide relevant results to substantiate this claim.

The statement regarding "high decoding requiring only 24 seconds of training data per word" is related to the length of the internal speech phase duration and the data incorporated into the decoder. In particular, the internal speech phase lasts for 1.5 seconds, with the relevant data used for decoder training. With 16 trials, each lasting 1.5 seconds, this accumulates to a total of 24 seconds of data integrated into the decoder, contributing to the successful outcomes observed in the best online internal speech trial results (91%). We clarified this statement in our manuscript.

Furthermore, high decoding was achievable with only 24 seconds of training data per word, corresponding to 16 trials x 1.5s of data.

b. The authors should cite previous literature to compare and contrast their findings regarding "strong evidence for internal speech processing at the single-neuron level in SMG."

We appreciate the suggestion to cite previous literature for the purpose of comparing and contrasting our findings related to "strong evidence for internal speech processing at the single-neuron level in SMG."

In our manuscript, we have conducted a comprehensive comparison with the most recent and relevant studies in the field, particularly those showcasing state-of-the-art internal speech decoding outcomes. We believe this thorough review of the current literature encompasses the necessary breadth for a comprehensive analysis. However, should there be any additional pertinent studies that we may have inadvertently overlooked, we are certainly open to including them to further bolster the robustness of our comparative analysis. We want to clarify that our comparative analysis was intentionally focused on "internal speech decoding" studies, distinct from investigations involving miming (silent speech) or attempted speech, as the novelty of our findings resides in this specific context. From our introduction:

In Pei, Barbour, et al., 2011 patients implanted with ECoG grids over frontal, parietal, and temporal regions silently read or vocalized written words from a screen. They significantly decoded vowels (37.5%) and consonants (36.3%) from internal speech (chance level 25%). Ikeda et al. 2014) decoded three internally spoken vowels using ECoG arrays using frequencies in the beta band, with up to 55.6% accuracy from Broca area (chance level 33%). Using the same recording technology, Martin et al., 2016 investigated the decoding of six words during internal speech. The authors demonstrated an average pair-wise classification accuracy of 58%, reaching 88% for the highest pair (chance level 50%). These studies were so-called open-loop experiments, in which the data was analyzed offline after acquisition. A recent paper demonstrated real-time (closed-loop) speech decoding using stereotactic depth electrodes (Angrick et al. 2021). Results were encouraging as internal speech could be detected; however, the reconstructed audio was not discernable and required audible speech to train the decoding model.

c. It is intriguing that the authors observed face and lip activity in a predominantly S1 arm region during vocalized speech. While it remains unclear if any language task was administered by Armenta Salas et al. (2018), I wonder if their fMRI data detected any significant activity in the S1 arm region during overt speech tasks.

We appreciate your observation. Our study did not involve a fMRI speech localizer task, consequently, our findings of face and lip activity in a predominantly S1 arm region during vocalized speech are not directly comparable to any fMRI data. Under methods:

Placement of electrodes was based on fMRI tasks involving grasp and dexterous hand movements.

d. The statement "However, in certain contexts, we observed identical or better classification accuracy during internal speech than vocalized speech (Figure 5A, Online decoding)" is not supported by Figure 5A, as it does not show online decoding. Additionally, Figure 5 does not indicate better classification accuracy during internal speech than vocalized speech. Please revise this statement to align with the available data.

Our intention behind the statement was to highlight the comparability of our online results to our offline vocalized speech outcomes. We acknowledge that this specific comparison may not provide sufficient evidence to assert superior internal speech decoding over vocalized speech decoding. We have removed the statement accordingly.

However, in certain contexts, we observed identical or better classification accuracy during internal speech than vocalized speech (Figure 5A, Online decoding)

Additionally, cross-phase decoding of vocalized speech from models trained on data during internal speech resulted in comparable classification accuracies to those of internal speech (Figure 5A,B, Internal). ~~However, in certain contexts we observed identical or better classification accuracy during internal speech than vocalized speech (Figure 4A, Online decoding).~~ Most neurons tuned during internal speech were also tuned to the same words during vocalized speech (53-56%, Figure 5C).

e. Similar to Participant 1, the authors should report the decoding accuracies for online and offline decoding in Participant 2 to provide a comprehensive analysis.

We acknowledge the need to report decoding accuracies for online and offline decoding in participant 2. Similar to Participant 1, we have now included the decoding accuracies for both online and offline decoding in the case of Participant 2, as well as a tuning analysis. Please see figures as response to comment b) ii) for the decoding analysis (Figure S4, panel B). Results for online analysis were added in Figure 5, attached below.

Figure 5 | Words can be significantly decoded during internal speech in SMG. A) Offline decoding accuracies: “Audio cue” and “Written cue” tasks data were combined for each individual session day, and leave one out cross-validation was performed (black dots). PCA was performed on the training data, a LDA model was constructed, and classification accuracies were plotted with 95% c.i, over the session means. Significance of classification accuracies was evaluated by comparing results to a shuffled distribution (averaged shuffle results = red dots, */**= average mean > 97.5 /99.5 percentile of shuffle distribution). In participant 1, classification accuracies during action phases (Cue, Internal, Speech)

following rest phases (ITI, D1, D2) were significantly higher (paired t-test: *** $p < 0.001$). **B)** Online decoding accuracies: Classification accuracies for internal speech were evaluated in a closed-loop internal speech BMI application on three different session days for both participants. In participant 1, decoding accuracies significantly improved (paired t-test, $p < 0.05$) when 16-20 trials per words were used to train the model, averaging 79% classification accuracy. In participant 2, online decoding accuracies were significant and averaged 23%. **C)** Offline confusion matrix for participant 1: Confusion matrices for each of the different task phases were computed on the tested data, and averaged over all session days. **D)** Online confusion matrix: A confusion matrix was computed combining all online runs, leading to a total of 304 trials (38 trials per word) for participant 1 and 448 online trials for participant 2. Participant 1 displayed comparable online decoding accuracies for all words, while participant 2 had preferential decoding for the words “Swimming” and “Spoon”.

5. Acknowledgements: To comply with patient privacy, please refrain from stating the participants' initial names (FG and AN) if they are not pseudonyms or identifiers.

The initials were removed from the acknowledgments.

Reviewer #3:

Remarks to the Author:

The authors present single unit and multi-unit activity data recorded from multi-electrode arrays chronically implanted in the left supramarginal gyrus (SMG) of 2 volunteer tetraplegic participants and additionally in the arm area of the left primary somatosensory cortex (S1) of 1 of the volunteer tetraplegic participants performing a single word language task. In the task, participants either hear a word in audio form (performed by 1 of the participants) or read a word in text form (performed by 1 of the participants), after which they are instructed to say the word internally first without vocalization in one time period, and the vocally say the word in a following time period after a brief delay preceding each period. The stimuli consist of 6 multi-syllable real English words and 2 multi-syllable pseudowords. The authors report that in the SMG of both patients single and multi-unit activity were activated for both internal and vocalized speech. In the patient who performed both the audio and written versions of the task, the responses in the word presentation time period differentiated between whether the word was presented in audio or text form, but the responses in both the internal and vocalized speech versions of the task did not differentiate between the modality of the word cue. In that patient, the SMG unit responses to different words were fed to both offline and online classification algorithms that were able to decode which of the 8 words were presented and repeated by the participant on a given trials with good success. This included successful decoding during the internal speech period, providing an

important proof-of-principle for potential future language brain-machine interface (BMI) applications in the SMG. Follow-up analyses suggested largely similar codes in terms of which units prefer which words between the internal and vocalized speech periods. In contrast, activity in the S1 of the same patient was only active during the speech vocalization time period, which the authors attribute to the existence of face and lip sensory activity that there is previous evidence of existing within the arm area of S1. An additional control experiment in one patient with a subset of words suggested that both a sound-based internal speech strategy and internalized visualization of the text representation of a word yielded similar success rates in decoding in the SMG.

This manuscript presents unique and important data for both the BMI and neuroscience of language fields. The inclusion of pseudo-words in the experimental stimuli provides some important insight suggesting that the language signals in the SMG during this task may likely be phonetic in nature. The cross-phase validation analyses that the authors employ show that the neural code in this area is largely preserved through-out word perception responses, internal speech and vocalized speech.

With that said, there are a number of issues to point out in the present manuscript, all of which are addressable.

We thank the reviewer for their in-depth review of our manuscript and for underlining the importance of our findings for the BMI and neuroscience community. Below, we have also addressed each comment point by point.

1. The authors should attach significance testing to their Figure 3 Demixed PCA analyses or otherwise show the temporal trajectory of significant word decoding identity in their data. For publication in Nature Human Behavior, that is not an unreasonable request, as similar articles in this space, including from this group (e.g. Aflalo et al. 2020) have done this. There are also issues with their execution and description of the Demixed PCA analysis, which I describe later on in these comments.

2. The authors mention that due to software constraints, the auditory cue appeared on average 250 ms later than the written cue, although the 2 single trial examples in Figure 2 suggest the difference is closer to ~750 ms. These things happen in the course of conducting experiments, but unfortunately it does complicate the inferences that can be gleaned from the cue period, which the present manuscript has not adequately addressed. The authors should account for this onset differences in their analyses

and in the figures showing the time course in the cue period (Figure 3A, Figure 3, Figure S2A). Ideally this should be done on a word-per-word basis, as altogether, this comes across as though there was some unintended variability in the onset time of the audio signal. With regards to this issue and Figure 3A, right now the most striking difference between the auditory and written cues in Figure 3A that a reader would notice is the latency difference between the two, which the authors don't discuss. So the issue needs to be addressed. The neural difference between the auditory and written cue responses show a latency difference that seems closer to the ~750 ms number from the examples in Figure 2.

Comments 1 and 2 were addressed in the same paragraph:

We agree that the temporal trajectory of significant word decoding is of interest and computed it following Aflalo's et al. 2020 method. Data were binned in 300ms, advancing by 50ms for each timepoint. Timepoints where decoding accuracies reached > 97.5 percentile of the shuffled distribution were marked on the plot.

We appreciate the reviewer's feedback regarding the concerns related to our dPCA analysis. We have taken these concerns seriously and have addressed them, as response to comment 8.

We concur with your observation regarding the alignment issue between the start of the auditory and written cues. Ideally, aligning each trial individually based on the onset of the auditory cue would be optimal. However, we emphasize that the primary focus of this paper lies in the decoding of internal and vocalized speech, rather than the explicit examination of auditory and written cue processing. For this reason, we do not draw conclusions regarding the onset and processing time of different cue modalities.

The dPCA analysis (Cue modality marginalization), along with the additional cross-decoding analysis we've included in response to Reviewer 1's feedback, underscores the cue-independent nature of internal and vocalized speech activity. This suggests that the alignment issue should not significantly impact our overall analysis.

The beforementioned ~250ms delay corresponds to the difference in time between the written cue appearing on the screen, and the start of the audio cue. By incorporating this adjusted timing, we have observed in the tuning analysis that the onset of tuning now aligns more closely for both audio and written cue trials. The figure was adapted in the manuscript (Figure 3 panel A).

Figure 3 | Neuronal population activity modulates for individual words. A) Average percentage of tuned neurons to words in 50ms time bins in SMG over the trial duration for “Auditory cue” (blue) and “Written cue” (green) tasks (solid line: mean over 10 sessions, shaded area: 95% confidence interval). **During the cue phase, neural data was shifted by 250ms, as the written cue appeared on average 250ms earlier on the screen than the auditory sound. B)** Average percentage of tuned neurons computed on firing rates per task phase, with 95% confidence interval over 10 sessions. Tuning during action phase (Cue, Internal, Speech) following rest phases (ITI, D1, D2) was significantly higher (paired t-test: ** $p < 0.01$, *** $p < 0.001$). **C)** Number of neurons tuned to each individual word in each phase for the “Auditory cue” and “Written cue” tasks. **D-F)** Same as A-C) for participant 2. **Due to reduced number of tuned units, only the written cue task variation was performed.**

Furthermore, the decoding analysis is performed on activity that is averaged over the entire cue phase duration. As can be seen in these example audio waveforms obtained from the cue phase, while there is some variability in the onset and offset of the different audio cue trials, activity is largely contained within the cue phase duration.

To conclude, we acknowledge that the difference in cue onset is undesirable, but it does not alter the findings and conclusions of our work.

3. There is some sloppiness to the writing. For example at the bottom of page 21, whatever study that sentence was supposed to reference is not there. The writing comes across as being written by a non-native English speaker. There are a number of examples of the wrong preposition being used, which is admittedly not easy to master in English. As one example of unclarity in the writing, in the last paragraph of page 10, for the sentence: "These results were significantly higher for internal speech and vocalized speech" It is not possible for a reader to determine what this refers to. Probably Cue, Internal, and Speech periods were compared to the ITI period, although in the context of this paragraph the text itself suggests Cue, Internal, and Speech conditions were compared to each other. The next version of the manuscript should be edited in that regard by a native English speaker before it's submitted.

The citation on page 21 has been clarified by placing it directly after the sentence in question. Additionally, the paper's English has been verified by a native speaker to ensure its accuracy and fluency.

During silent reading of a cued sentence with a neutral vs. increased prosody (madeleine brought me vs. MADELEINE brought me), one study in particular found increased left SMG activation correlated with the intensity of the produced inner speech (Lœvenbruck et al. 2005).

4. It would be considerably helpful if the authors could include single neuron examples of neuronal activity traces that discriminate between words, to give the reader a better sense of what is driving the word decoding results.

We added supplementary Figure S2 in the manuscript, showing neuronal responses to all words for Participant 1 and Participant 2, as can be seen in response to reviewer 1, comment 7. Specifically, columns 1 and 2 show the same unit's response to Auditory cue trials and Written cue trials on the same session day.

5. I suspect the following might not have been done due to sensitivity concerns, but it bears mentioning here that including the location of the implants in Figure 2 rather than Figure S4B would be helpful scientifically if the authors are able to do it. The precise location of these implants are of course of great interest to the scientists interacting with this work.

We agree that the implant locations are of great interest to the scientific community. We added the precise locations in Figure 1, please see the figure attached as a response to comment 2.i.a from reviewer number 2.

6. I was amazed that the text in the article does not mention anywhere that the array is in the left hemisphere. This detail is important for all of the implications of this study. As it stands, one can only gather that this from the orientation of the brain in Figure S4B.

We acknowledge the oversight in not specifying the location on the left side and added that information in the manuscript.

Data were collected from implants located in the **left** supramarginal gyrus (SMG) and the **left** primary somatosensory cortex (S1).

7. The authors' use of the word "tuning" in this manuscript does not match the analyses that they conducted. For example, in the visual system, to say that a neuron is "tuned" to orientation means that it responds preferentially to bars of light of certain orientations but not of other orientations. In this manuscript, tuning for words would reflect significant preference for one word over another word. However in the analysis the authors conducted, a unit was considered tuned if its response to at least one word was significantly larger than its ITI response. Thus a neurons that responded to every word equally would be considered tuned. The authors should change either the label they use or their analyses to assess tuning. For example a significant ANOVA or Kruskal-Wallis test across words would show significant word-tuning as the word is typically used.

We acknowledge the inherent limitation of the linear regression model potentially considering neurons responding equally to all words as tuned. To address this, we have performed the Kruskal-Wallis analysis, the results of which were added as supplementary Figure S3 panel C and D in the manuscript. Below as a response to the reviewer is the direct comparison between tuning defined through the linear regression analysis (Figure 3 panel B and E) and the Kruskal-Wallis analysis (Figure S3, panel C,D). Notably, both analyses in participant 1 yield comparable outcomes, reinforcing the consistency and reliability of our findings.

Participant 1:

Participant 2:

8. In the authors' Demixed PCA analysis, they first state that they followed the method described in Kobak et al. 2016, but the details that they describe after that differs from what Kobak et al. 2016 describe, in a manner such that the way that the what the present authors did differs from what Kobak et al. 2016 did does not appear to be sensible. Briefly, the 8 terms of the ANOVA-like decomposition in Kobak et al. 2016 are grouped into 5 terms by grouping e.g. the stimulus term with the stimulus-time interaction term, with the time term appearing alone. Here the authors apparently combined the time term with cue modality-word-time interaction to get their ultimate "Timing" factor, which makes no sense. If the authors follow Kobak et al. 2016 they would have a fourth type of component for cue modality-word interactions. Following from the confusion with the analysis, the authors' description of this analysis in the methods is not self-consistent, so care is needed both for this analysis itself and the descriptions of this analysis in the next version of the manuscript. Some other details to point out, the authors state that individual components are synonymous with marginalizations, which is not correct. They state that ANOVA involves covariance decomposition when it involves variance decomposition.

Finally, it would be helpful for the reader unfamiliar with the technique to specify in the results text and caption that Figure 3 shows the projection of the data onto the respective dPCA decoder axis.

We appreciate your feedback, and we have taken steps to enhance the clarity and alignment of our dPCA description and analysis with the framework outlined by Kobak et al. (2016). Thank you for your constructive guidance in optimizing the presentation of our work. The revised method section can be found as response to reviewer 1, under comment 5 and 6. The updated figure can be found as response to reviewer 2, under 2b.ii).

Additionally, we added the following text in the manuscript under results:

In Figure 3, demixed principal components explaining the highest amount of variance were plotted by projecting data onto their respective dPCA decoder axis.

We added the following text in the manuscript in the figure legend:

Demixed principal components explaining the highest variance within each marginalization were plotted over time, by projecting the data onto their respective dPCA decoder axis.

9. It is unnecessarily difficult for the reader to discern which results are from which participant. The authors should either add a blanket statement in the beginning of the methods that “unless indicated otherwise, results are for patient 1” or say individually for each figure e.g. “this shows data from participant 1”. It is also strange that the Kruskal-Wallis analysis was done on participant 2’s data, but not participant 1’s data. Ideally a Kruskal-Wallis analysis of participant 1’s data would be a better estimate of tuning in a neuron rather than the t-test that the authors employed. Additionally, the inclusion of both participant 2’s data and the S1 data both strengthen this manuscript overall, however the reader is left to wonder why the S1 data from Patient 2 was not analyzed. The authors should either include analyses of these data, or explain why they are not doing that.

We agree with the reviewer's suggestion regarding the inclusion of Participant 2's data figures and their corresponding S1 data. In response, we have recorded additional data, and generated analogous figures to those of Participant 1, focusing for the decoding and tuning analyses. We have observed a

limited presence of tuned neurons for Participant 2, resulting in relatively lower classification accuracies. Additionally, the S1 results for Participant 2 align with those of Participant 1. Furthermore, we have labeled Figures in more detail, focusing on which Figure corresponds to which participant, to increase the clarity of the manuscript. Figure 2-5 in the manuscript were adapted, and Figure S2,S3 and S4 were added. These changes serve to clarify our observations and contribute to a more comprehensive understanding of our research.

Figure 3 | Neuronal population activity modulates for individual words. A) Average percentage of tuned neurons for participant 1 to words in 50ms time bins in SMG over the trial duration for “Auditory cue” (blue) and “Written cue” (green) tasks (solid line: mean over 10 sessions, shaded area: 95% confidence interval). The written cue appeared on average 250ms earlier on the screen than the auditory sound. **B)** Average percentage of tuned neurons for participant 2 computed on firing rates per

task phase, with 95% confidence interval over 10 sessions. Tuning during action phase (Cue, Internal, Speech) following rest phases (ITI, D1, D2) was significantly higher (paired t-test: ** $p < 0.01$, *** $p < 0.001$). **C)** Number of neurons tuned for participant 1 to each individual word in each phase for the “Auditory cue” and “Written cue” tasks. **D-F) Same as A-C) for participant 2. Due to reduced number of tuned units, only the written cue task variation was performed.**

combining all online runs, leading to a total of 304 trials (38 trials per word) for participant 1 and 448 online trials for participant 2. **Participant 1 displayed comparable online decoding accuracies for all words, while participant 2 had preferential decoding for the words “Swimming” and “Spoon”.**

Figure S4 | S1 shows generalized word activity during vocalized speech. A) Participant 1 S1 decoding analysis. “Auditory cue” and “Written cue” tasks data were combined for each individual session day, and leave one out cross-validation was performed (black dots). PCA was performed on the training data, a LDA model was constructed, and results were plotted with 95% c.i. of the session means. Significance of classification accuracies was evaluated by comparing results to a shuffled distribution (averaged

shuffle results = red dots). No classification accuracy was significant. However, classification accuracy during vocalized speech was significantly higher than during the previous delay period (paired t-test: $*p < 0.05$). Lack of decoding during the cue phase suggest no auditory contamination occurred in S1 channels. B) C) Same as A) but for participant 2.

10. The introduction would benefit from a paragraph motivating the author's decision to include pseudowords in their stimuli. Similarly they should in the introduction briefly motivate the reason to include S1. The authors mention that they hypothesize that S1 would modulate in the speech period only, but there is no explanation at all of what that hypothesis is based on in the introduction, although it does come in the discussion section later.

The inclusion of pseudowords allows us to probe whether semantic comprehension is a required for SMG's word representation. The following sentence was added to the introduction:

In this work, two participants with tetraplegia performed internal and vocalized speech of eight words while neurophysiological responses were captured from two implant sites. **To investigate neural semantic and phonetic representation, the words were composed of six lexical words and two pseudowords (words that mimic real words without semantic meaning).**

We added a sentence in the introduction explaining our motivation behind our hypothesis for S1 modulation during speech.

S1 was included as a control for movement due to emerging evidence of its activation beyond defined regions of interest (Muret et al. 2022; Rosenthal et al. 2023).

11. In the linear regression analysis section of the methods, there is a “++” typo in the equation, and I would recommend subscripts for the numbers rather than “B1X1” if that equation is ultimately included. However, I don't think the description of this as a linear regression is appropriate- as the authors' text describes, what they performed are 8 t-tests comparing the activity to each ITI period. This would be more clearly described as 8 t-tests with a pooled ITI baseline term. Moreover the t-tests used are describe as 'student's t-tests' which generally refer to unpaired, 2 sample t-tests. Paired t-tests are the correct tests to use in this instance.

The alpha level for the Kruskal-Wallis test applied to participant 2's data is not mentioned, which

becomes particularly important detail as the authors find proportions near %5 with significant results, which they state reflects a significant proportion of their recorded population. They authors need to state the alpha they used for this test, and follow it up with a binomial test in order to conclude that there are a significant proportion of tuned neurons in their population.

We have adapted our wording for the method description of tuning defined by linear regression analysis.

To identify units exhibiting selective firing rate patterns (or tuning) for each of the eight words, linear regression analysis was performed in two different ways: 1) step by step in 50ms time bins to allow assessing changes in neuronal tuning over the entire trial duration; 2) averaging the firing rate in each task phase to compare tuning between phases. The model returns a fit that estimates the firing rate of a unit based on the following variables:

$$FR = \sum_{w=1}^W \beta_w X_w + \beta_0$$

where FR corresponds to the firing rate of the unit, β_0 to the offset term equal to the average ITI data of the unit, **X is the vector indicator variable for each word w** , and β_w corresponds to the estimated regression coefficient for word w . W was equal to 8 (Battlefield, Cowboy, Python, Spoon, Swimming, Telephone, Bindip, Nifzig). (see Wandelt et al. 2022).

In this model, β symbolizes the change of firing rate from baseline for each word. **A t statistic was calculated by dividing each β coefficient by its standard error. Tuning was based on the p-value of the t-statistic for each beta coefficient.**

Furthermore, we have calculated the outcomes of the Kruskal-Wallis test for participant 2. A unit was defined as tuned when the fdr corrected p-value < 0.05. The 95% confidence interval of the mean over 9 session days was calculated. The figure has been added as Figure S3, panel D in the manuscript. As the confidence interval of the internal speech phase does not include zero, and is significantly higher than tuning during the ITI phase (paired t-test, $p = 0.016$), we conclude that the number of tuned units during the internal speech phase is significantly different than baseline activity

D

Kruskal-Wallis tuning - SMG

12. The details of the authors' classification analyses are under-specified. They describe using a training and testing set, but don't mention how they partitioned the data into these 2 sets, aside from saying that they performed leave one out validation. This language is unnecessarily unclear- they should just say that they used a leave one out method. They should describe what the one item left out was – one neurons vs. one trial across neurons vs. one trial for one neuron. For the cross-phase classification, the authors' say they performed that following the details of the intra-phase classification, however how that would occur is not at all clear- they need to specify what was left out in the cross-phase classification and tested across iterations. The alpha level of 0.001 for the test is unusually low, which if used requires explanation.

We incorporated the reviewers feedback and reworked the description of the classification accuracy.

Using the neuronal firing rates recorded during the tasks, a classifier was used to evaluate how well the set of words could be differentiated during each phase. Classifiers were trained using averaged firing rates over each task phase, resulting in 6 matrices of size n,m , where n corresponds to the number of trials, and m corresponds to the number of recorded units. A model for each phase was built using linear discriminant analysis (LDA), assuming an identical covariance matrix for each word, which resulted in best classification accuracies. Leave-one-out cross-validation was performed to estimate decoding performance, leaving out a different trial across neurons at each loop. Principal component analysis (PCA) was applied on the training data and PCs explaining more than 95% of the variance were selected

as features and applied to the single testing trial. A 95% confidence interval was computed as described above.

The alpha value of the cross-phase classification was replaced by an *fdr* corrected *p*-value (Benjamini and Hochberg 1995; Benjamini and Yekutieli 2001) by number of tests per phase, with an alpha of 0.05 (Figure 6, panel A and B).

Shared neural representation between language processes - Participant 1

13. There are some unspecified details that should be added to the methods section. The authors should say in the methods how many trials were in the session, and the randomization and balancing procedure used, and the order in which the tasks were performed (auditory cue vs. written cue) in participant 1 in a given day.

On each session day, 2 blocks of experiments were performed, one block cueing words with an auditory cue, and one block cueing words with a written cue. In each block, 8 trials of each word were performed, leading to 64 trials per block (minus potential error trials). On four (nonconsecutive)

session days, the auditory condition was run first and on six (nonconsecutive) session days, the written cue condition was run first.

The following text was added in the method section:

Total number of recording trials

For participant 1, we collected offline datasets composed of 8 trials per word for each cue across ten sessions. Trials where errors occurred were excluded. In total, between 156-159 trials per word were included, leading to a total of 1257 trials for offline analysis. On four nonconsecutive session days, the Auditory cue task was run first, and on six nonconsecutive days, the Written cue task was run first. For online analysis, datasets were recorded on three different session days, for a total of 304 trials. Participant 2 underwent a similar data collection process, with offline datasets comprising 16 trials per word using the written cue modality over nine sessions. Error trials were excluded. In total, between 142-144 trials per word were kept, leading to a total of 1145 trials for offline analysis. For online analysis, datasets were recorded on three session day, leading to a total of 448 online trials.

14. For Figure 6, for testing the proportions of cells with a binary feature like tuned vs. not-tuned, Fisher's exact test would be the appropriate test. In the last sentence of page 17, the authors seem to be describing a comparison between the second-most right bars in the plots of 5B versus the same in 5A. However that comparison does not seem like it would be significant. Please clarify this.

We appreciate that the clarity in the text could be improved. In the text, we compare the cross-decoding accuracies within (Figure 6A,B, x axis = Speech) plots, rather than comparing the second-most right bar of both plots. The written cue phase and the internal cue phase can be decoded to similar extends from vocalized speech (Figure 6B, Speech), meaning that there is no significant decoding difference between the accuracy for Internal and Written Cue. However, the auditory cue is cross-decoded significantly lower than internal speech from vocalized speech (Figure 6A, Speech), demonstrated by significantly lower decoding of the Auditory cue phase compared to the Internal phase.

The manuscript was adapted as following.

A model trained on vocalized speech demonstrated an equally strong ability to generalize between internal speech and written words (Figure 6B, Speech), as evidenced by similar decoding accuracy between the Internal and Cue. However, the model generalized significantly better to internal speech than the representation found during the auditory cue (Figure 6A, Speech), as the Internal phase was decoded significantly higher than the Cue phase.

- Angrick, Miguel, Maarten C. Ottenhoff, Lorenz Diener, Darius Ivucic, Gabriel Ivucic, Sophocles Goulis, Jeremy Saal, et al. 2021. "Real-Time Synthesis of Imagined Speech Processes from Minimally Invasive Recordings of Neural Activity." *Communications Biology* 4 (1): 1–10. <https://doi.org/10.1038/s42003-021-02578-0>.
- Gajardo-Vidal, Andrea, Diego L. Lorca-Puls, Ploras Team, Holly Warner, Bawan Pshdary, Jennifer T. Crinion, Alexander P. Leff, et al. 2021. "Damage to Broca's Area Does Not Contribute to Long-Term Speech Production Outcome after Stroke." *Brain: A Journal of Neurology* 144 (3): 817–32. <https://doi.org/10.1093/brain/awaa460>.
- Guenther, Frank H., Jonathan S. Brumberg, E. Joseph Wright, Alfonso Nieto-Castanon, Jason A. Tourville, Mikhail Panko, Robert Law, et al. 2009. "A Wireless Brain-Machine Interface for Real-Time Speech Synthesis." *PLOS ONE* 4 (12): e8218. <https://doi.org/10.1371/journal.pone.0008218>.
- Ikeda, Shigeyuki, Tomohiro Shibata, Naoki Nakano, Rieko Okada, Naohiro Tsuyuguchi, Kazushi Ikeda, and Amami Kato. 2014. "Neural Decoding of Single Vowels during Covert Articulation Using Electrocorticography." *Frontiers in Human Neuroscience* 8. <https://www.frontiersin.org/articles/10.3389/fnhum.2014.00125>.
- Lœvenbruck, H el ene, Monica Baci u, Christoph Segebarth, and Christian Abry. 2005. "The Left Inferior Frontal Gyrus under Focus: An fMRI Study of the Production of Deixis via Syntactic Extraction and Prosodic Focus." *Journal of Neurolinguistics* 18 (3): 237–58. <https://doi.org/10.1016/j.jneuroling.2004.12.002>.
- Martin, Stephanie, Peter Brunner, I naki Iturrate, Jos e del R. Mill an, Gerwin Schalk, Robert T. Knight, and Brian N. Pasley. 2016. "Word Pair Classification during Imagined Speech Using Direct Brain Recordings." *Scientific Reports* 6 (1): 25803. <https://doi.org/10.1038/srep25803>.
- Pei, Xiaomei, Dennis L. Barbour, Eric C. Leuthardt, and Gerwin Schalk. 2011. "Decoding Vowels and Consonants in Spoken and Imagined Words Using Electrocorticographic Signals in Humans." *Journal of Neural Engineering* 8 (4): 046028. <https://doi.org/10.1088/1741-2560/8/4/046028>.
- Wandelt, Sarah K., Spencer Kellis, David A. Bj anes, Kelsie Pejsa, Brian Lee, Charles Liu, and Richard A. Andersen. 2022. "Decoding Grasp and Speech Signals from the Cortical Grasp Circuit in a Tetraplegic Human." *Neuron*, March. <https://doi.org/10.1016/j.neuron.2022.03.009>.
- Willett, Francis R., Erin M. Kunz, Chaofei Fan, Donald T. Avansino, Guy H. Wilson, Eun Young Choi, Foram Kamdar, et al. 2023. "A High-Performance Speech Neuroprosthesis." *Nature* 620 (7976): 1031–36. <https://doi.org/10.1038/s41586-023-06377-x>.

Decision Letter, first revision:

8th January 2024

Dear Dr. Wandelt,

Thank you for your patience as we've prepared the guidelines for final submission of your Nature Human Behaviour manuscript, "Representation of internal speech by single neurons in human supramarginal gyrus" (NATHUMBEHAV-23051452A). Please carefully follow the step-by-step instructions provided in the attached file, and add a response in each row of the table to indicate the changes that you have made. Please also address the additional marked-up edits we have proposed within the reporting summary. Ensuring that each point is addressed will help to ensure that your revised manuscript can be swiftly handed over to our production team.

We would hope to receive your revised paper, with all of the requested files and forms within two-three weeks. Please get in contact with us if you anticipate delays.

Nature Human Behaviour offers a Transparent Peer Review option for new original research manuscripts submitted after December 1st, 2019. As part of this initiative, we encourage our authors to support increased transparency into the peer review process by agreeing to have the reviewer comments, author rebuttal letters, and editorial decision letters published as a Supplementary item. When you submit your final files please clearly state in your cover letter whether or not you would like to participate in this initiative. Please note that failure to state your preference will result in delays in accepting your manuscript for publication.

In recognition of the time and expertise our reviewers provide to Nature Human Behaviour's editorial process, we would like to formally acknowledge their contribution to the external peer review of your manuscript entitled "Representation of internal speech by single neurons in human supramarginal gyrus". For those reviewers who give their assent, we will be publishing their names alongside the published article.

Cover suggestions

We welcome submissions of artwork for consideration for our cover. For more information, please see our guide for cover artwork.

ORCID

Non-corresponding authors do not have to link their ORCIDs but are encouraged to do so. Please note that it will not be possible to add/modify ORCIDs at proof. Thus, please let your co-authors know that if they wish to have their ORCID added to the paper they must follow the procedure described in the following link prior to acceptance: <https://www.springernature.com/gp/researchers/orcid/orcid-for-nature-research>

Nature Human Behaviour has now transitioned to a unified Rights Collection system which will allow our Author Services team to quickly and easily collect the rights and permissions required to publish your work. Approximately 10 days after your paper is formally accepted, you will receive an email in providing you with a link to complete the grant of rights. If your paper is eligible for Open Access, our Author Services team will also be in touch regarding any additional information that may be required to arrange payment for your article.

Please note that *Nature Human Behaviour* is a Transformative Journal (TJ). Authors may publish their research with us through the traditional subscription access route or make their paper immediately open access through payment of an article-processing charge (APC). Authors will not be required to make a final decision about access to their article until it has been accepted. Find out more about Transformative Journals

[REDACTED]

Best regards,
Alex McKay
Editorial Assistant
Nature Human Behaviour

On behalf of

Giacomo Ariani
Editor
Nature Human Behaviour

Reviewer #1:

Remarks to the Author:

The authors have thoroughly answered all my comments in a convincing way. I acknowledge their corresponding additional work and congratulate them for this nice study that I recommend for publication.

Reviewer #2:

Remarks to the Author:

I appreciate the authors for their comprehensive revision of the manuscript in response to my initial review. They have invested considerable effort in addressing the raised points, resulting in a significantly improved manuscript.

Although I appreciate and respect the authors' responses in the authors' reply, I would like to highlight some minor points of disagreement. I believe these discrepancies are minor in nature and should not adversely affect the overall quality of the manuscript.

I extend my congratulations to the authors for their exemplary work in crafting this manuscript.

Reviewer #3:

Remarks to the Author:

The authors have improved the manuscript considerably, and from my point of view have adequately responded to most of the concerns in the review.

There are some remaining and additional concerns I have after going through the revised manuscript that bear mentioning here.

The discussion in the lines from 475 to 484 that compares decoding in the cue period between the auditory and visual conditions weakens the overall manuscript. The significance in a non-statistical sense of the comparison that the authors make is unclear, stemming from the problems from the visual vs. auditory stimulus alignment that the authors have. The authors' updated Figure 3 does the bare minimum of accounting for their reported mean 250 ms onset difference between the Aud vs. Vis conditions, but none of their other analyses through-out this manuscript accounts for that. The analyses that the authors have done cannot disentangle potential differences in the strength of decoding between cue modalities from the latency differences due to the onset problem and the fact

that the visual cue condition thus has a longer time period of relevant neural data for the decoding, while the audio condition has more of what is effectively noise added to it before the stimulus starts that would degrade the decoding. Even with correctly aligning the stimulus onsets on every trial it is difficult to make the apples-to-oranges comparison between visual and auditory language stimuli where the visual text appears nearly instantaneously but the auditory stimulus takes a few hundreds of milliseconds to complete, although it is noteworthy that the authors do not even have that correct stimulus alignment, either by choice or because they are unable to do it.

A line of reasoning that the authors make elsewhere in their rebuttal letter is that the focus of this manuscript is the decoding in the internal phase of the trial, and indeed that line of reasoning is sensible. What is very convincing in Figure 5 is that everything from the internal phase onward in the trial is remarkably similar between the auditory and written cue conditions, which is a meaningful result in itself. The manuscript would be better served if the authors applied their approach from their rebuttal letter to these paragraphs of the manuscript and tempered the inferences from the written versus auditory modalities in the cue phase rather than emphasized them. Or the authors could conduct convincing analyses to show what they are claiming they have shown, but the present analyses are not convincing in the ways that they are described in the article text.

Moreover, the paragraph in blue from lines 480 to 484 remains to be poorly written and needlessly unclear. The authors description in their answer to my comment in their rebuttal letter is clear; this paragraph in the article text is not. It is needlessly difficult for a reader to tell what specific analyses are being referred to in this paragraph.

The comparison of the “sound imagination” and “visual imagination” in Figure S5 is a useful contribution to the manuscript even if the results of the manipulation were not significant. However, the manuscript would benefit if the text spelled out to the reader the specific conditions that were intended to be compared to reach the intended conclusion, and then actually reporting the outcomes of the hypothesis tests applied to the data, even if the outcomes of those tests were not significant.

In Figure 4 it would be helpful to add a description that the all of the panels, e.g. for Participant 1, plot 16 traces in each panel, where in panels A and B the line styles reflect cue modality-only while in panel C the line styles reflect individual word and cue modality; and then likewise for the Participant 2 panels.

In the sentence on lines 348-349: The authors can of course define the word “tuning” as it applies to this manuscript in the way that they prefer, but the wording of “we computed selectivity for individual words from the average FR in each task phase (Figure 3B,E)” is factually incorrect as to what it purports the analyses of Figure 3B,E to show and it should be changed. It would also be somewhat misleading to use the phrase “selectivity for individual words” to refer to the Kruskal-Wallis analyses, although that would be closer to correct. “Selectivity for individual words” means having a higher firing rate for specific words compared to other words.

Author Rebuttal, first revision:

Reviewer #1:

Remarks to the Author:

The authors have thoroughly answered all my comments in a convincing way. I acknowledge their corresponding additional work and congratulate them for this nice study that I recommend for publication.

We thank the reviewer for their valuable feedback and for their congratulations!

Reviewer #2:

Remarks to the Author:

I appreciate the authors for their comprehensive revision of the manuscript in response to my initial review. They have invested considerable effort in addressing the raised points, resulting in a significantly improved manuscript.

Although I appreciate and respect the authors' responses in the authors' reply, I would like to highlight some minor points of disagreement. I believe these discrepancies are minor in nature and should not adversely affect the overall quality of the manuscript.

I extend my congratulations to the authors for their exemplary work in crafting this manuscript.

We thank the reviewer for their valuable feedback and for their congratulations!

Reviewer #3:

Remarks to the Author:

The authors have improved the manuscript considerably, and from my point of view have adequately responded to most of the concerns in the review.

There are some remaining and additional concerns I have after going through the revised manuscript that bear mentioning here.

The discussion in the lines from 475 to 484 that compares decoding in the cue period between the auditory and visual conditions weakens the overall manuscript. The significance in a non-statistical sense of the comparison that the authors make is unclear, stemming from the problems from the visual vs. auditory stimulus alignment that the authors have. The authors' updated Figure 3 does the bare minimum of accounting for their reported mean 250 ms onset difference between the Aud vs. Vis conditions, but none of their other analyses through-out this manuscript accounts for that. The analyses that the authors have done cannot disentangle

potential differences in the strength of decoding between cue modalities from the latency differences due to the onset problem and the fact that the visual cue condition thus has a longer time period of relevant neural data for the decoding, while the audio condition has more of what is effectively noise added to it before the stimulus starts that would degrade the decoding. Even with correctly aligning the stimulus onsets on every trial it is difficult to make the apples-to-oranges comparison between visual and auditory language stimuli where the visual text appears nearly instantaneously but the auditory stimulus takes a few hundreds of milliseconds to complete, although it is noteworthy that the authors do not even have that correct stimulus alignment, either by choice or because they are unable to do it.

We recognize the noted discrepancies in the timing between the auditory and the written cue stimuli. Specifically, the latency in the auditory output ranged from 200ms to 650ms following the initiation of the cue phase. This inconsistency in delay was attributed to the utilization of varied sound outputs (direct computer audio and Bluetooth speaker). To rectify this issue in Figure 3A, we aligned the neural data with the onset of the audio for each trial and have revised the figure caption to reflect this adjustment. The updated figure suggests that neural activation for written and auditory cue processing is of comparable length.

Figure 3A: Average percentage of tuned neurons to words in 50ms time bins in SMG over the trial duration for “Auditory cue” (blue) and “Written cue” (green) tasks (solid line: mean over 10 sessions, shaded area: 95% confidence interval). **During the cue phase of auditory trials, neural data was aligned to audio onset, which occurred within 200-650ms following initiation of the cue phase.**

We maintain our stance that our methodology, which involves averaging activity over the entire cue phases (including tuning analysis and decoding analysis) is adequate to accurately compare SMG representation across auditory processing, written word recognition, internal speech and vocalized speech. To provide evidence for our claim, we aimed to demonstrate

that higher encoding strength of written cue processing is observed when neural data is aligned to auditory onset as well.

In that aim, we identified data subsets corresponding to peak neural activity. We then averaged the firing rate over these peak periods, and performed eight-word classification. The timepoints indicating start and end of the peak period are marked in the Figure below by vertical lines.

We compared peak classification accuracies (Figure B, Aligned), to decoding over 1.5 seconds of cue phase data (Figure B, Not Aligned). The latter corresponds to the methodology applied in the manuscript, specifically in Figure 6A. As expected, we observe an increase in decoding strength when accounting for cue onset. However, this increase in classification accuracy is observed both for the written cue dataset and the auditory cue dataset. Additionally, word classification during written cue is significantly higher than during auditory cue, both in the aligned and non-aligned datasets. This finding corroborates higher decoding of the written cue data emphasized in Figure 6A.

Based on these findings, we wish to reaffirm our decision to not align the neural data with cue onset, and to instead average the data over the 1.5s duration. Indeed, the aim of Figure 6 is to compare activity during internal speech to activity during cue and vocalized speech phases. Given the lack of observable behavioral output during internal speech, accurate alignment of neural activity during that experimental phase is not feasible. We hypothesize that aligning internal speech data, if it were possible, would similarly enhance decoding accuracies. Therefore, to ensure the most equitable comparison across different modalities, we opted to

average the neural activity over 1.5 seconds in each task phase. We are confident that our approach offers a fair comparison between task phases, and that our results in Figure 6 are robust.

A line of reasoning that the authors make elsewhere in their rebuttal letter is that the focus of this manuscript is the decoding in the internal phase of the trial, and indeed that line of reasoning is sensible. What is very convincing in Figure 5 is that everything from the internal phase onward in the trial is remarkably similar between the auditory and written cue conditions, which is a meaningful result in itself. The manuscript would be better served if the authors applied their approach from their rebuttal letter to these paragraphs of the manuscript and tempered the inferences from the written versus auditory modalities in the cue phase rather than emphasized them. Or the authors could conduct convincing analyses to show what they are claiming they have shown, but the present analyses are not convincing in the ways that they are described in the article text.

We believe that our findings presented in the manuscript are robust, whether we align the neural data to the auditory onset or utilize the entire 1.5 seconds of data for our decoding analysis (see figure in above paragraph). Nonetheless, we have included an additional sentence in our manuscript acknowledging the possible influence of processing time on our findings (Lines 246–248),

The strongest shared neural representations were found between visual word recognition, internal speech, and vocalized speech (Figure 6B). A model trained on internal speech was highly generalizable to both vocalized speech and written cued words, evidence for a possible shared neural code (Figure 6B, Internal). In contrast, the model’s performance was significantly lower when tested on data recorded in the auditory cue phase (Figure 6A, Internal, paired t-test, $p < 0.001$). These differences could stem from the inherent challenges in comparing visual and auditory language stimuli, which differ in processing time - instantaneous for text versus several hundred milliseconds for auditory stimuli.

Moreover, the paragraph in blue from lines 480 to 484 remains to be poorly written and needlessly unclear. The authors description in their answer to my comment in their rebuttal letter is clear; this paragraph in the article text is not. It is needlessly difficult for a reader to tell what specific analyses are being referred to in this paragraph.

We updated the paragraph to increase clarity. (Lines 254 – 260)

We evaluated the capability of a classification model, initially trained to distinguish words during vocalized speech, in its ability to generalize to Internal and Cue phases (Figure 6 A,B, x axis = Speech). The model demonstrated similar levels of generalization during internal speech and in response to written cues, as indicated by the lack of significance in decoding accuracy between the Internal and Written Cue phase (Figure 6B, x axis = Speech, Cue- Internal). However, the model generalized significantly better to internal speech than to representations observed during the auditory cue phase (Figure 6A,x axis = Speech, Cue -. Internal, $p < 0.001$).

The comparison of the “sound imagination” and “visual imagination” in Figure S5 is a useful contribution to the manuscript even if the results of the manipulation were not significant. However, the manuscript would benefit if the text spelled out to the reader the specific conditions that were intended to be compared to reach the intended conclusion, and then actually reporting the outcomes of the hypothesis tests applied to the data, even if the outcomes of those tests were not significant.

We enhanced the clarity of the text to help the readers to come to the intended conclusion. (Lines 286 – 293)

We therefore compared two possible internal sensory strategies: a “sound imagination” strategy in which the participant imagined hearing the word, and a “visual imagination” strategy in which the participant visualized the word’s image (Figure S5A). Each strategy was cued by the modalities we had previously tested (auditory and written words). To assess the similarity of these internal speech processes to other task phases, we conducted a cross-phase decoding analysis (as performed in Figure 6). We hypothesized that if the high cross-decoding results between internal and written cue phases primarily stemmed from the participant engaging in visual word imagination, we would observe lower decoding accuracies during the auditory imagination phase.

Given that this version of the experiment was performed during a single session, we refrained from performing statistical comparisons between conditions. However, as we were able to decode both the visual and the auditory imagination highly above chance (25%) (in contrast to chance level classification during the auditory imagination condition), we conclude that the findings presented in Figures 5 and 6 are not solely attributable to the participant engaging in visual imagination during the internal speech phase.

In Figure 4 it would be helpful to add a description that the all of the panels, e.g. for Participant 1, plot 16 traces in each panel, where in panels A and B the line styles reflect cue modality-only

while in panel C the line styles reflect individual word and cue modality; and then likewise for the Participant 2 panels.

Trace descriptions have been added to the figure's legend, to enhance understanding of the dPCA figure.

Figure 4 | Demixed principal component analysis (dPCA) highlights SMG's involvement in language processing. dPCA was performed to investigate variance within three marginalizations: "Timing", "Cue Modality", and "Word" for participant 1 (A-C) and "Timing" and "Word" for participant 2 (D-E). Demixed principal components explaining the highest variance within each marginalization were plotted over time, by projecting the data onto their respective dPCA decoder axis. **A)** The "Timing" marginalization demonstrates SMG modulation during cue, internal speech and vocalized speech, while S1 only represents vocalized speech. Blue solid lines (8) represent Auditory cue trial, green dashed lines (8) represent Written cue trials. **B)** The "Cue Modality" marginalization suggests internal and vocalized speech representation in SMG are not affected by the cue modality. Blue solid lines (8) represent Auditory cue trial, green dashed lines (8) represent Written cue trials. **C)** The "Word" marginalization shows high variability for different words in SMG, but near zero for S1. Colors (8) represent individual words. For each color, solid lines represent Auditory trials, dashed lines represent Written cue trials. **D)** Same as A) for participant 2. Green dashed lines (8) represent Written cue trials. **E)** Same as C) for participant 2. Colors (8) represent individual words during Written cue trials. Variance for different words in SMG (left) was higher than in S1 (right), but lower in comparison to SMG in participant 1 (C).

In the sentence on lines 348–349: The authors can of course define the word "tuning" as it applies to this manuscript in the way that they prefer, but the wording of "we computed selectivity for individual words from the average FR in each task phase (Figure 3B,E)" is factually incorrect as to what it purports the analyses of Figure 3B,E to show and it should be changed. It would also be somewhat misleading to use the phrase "selectivity for individual words" to refer to the Kruskal-Wallis analyses, although that would be closer to correct. "Selectivity for individual words" means having a higher firing rate for specific words compared to other words.

Thank you for your feedback on the wording in lines 348–349. Based on your clarification, the phrase "we computed selectivity for individual words from the average FR in each task phase (Figure 3B,E)" was rephrased accordingly: (Line 152 - 154)

To quantitatively compare activity between phases, ~~we computed selectivity for individual words from the average FR in each task phase~~ we assessed the differential response patterns for individual words by examining the variations in average firing rate across different task phases (**Figure 3B,E**).

Final Decision Letter:

Dear Dr Wandelt,

We are pleased to inform you that your Article "Representation of internal speech by single neurons in human supramarginal gyrus", has now been accepted for publication in *Nature Human Behaviour*.

Please note that *Nature Human Behaviour* is a Transformative Journal (TJ). Authors may publish their research with us through the traditional subscription access route or make their paper immediately open access through payment of an article-processing charge (APC). Authors will not be required to make a final decision about access to their article until it has been accepted. Find out more about Transformative Journals

With best regards,

Giacomo Ariani
Editor
Nature Human Behaviour